# Two subtypes of GTPase-activating proteins coordinate tip growth and cell size regulation in *Physcomitrium patens*

Jingtong Ruan[1,3], Linyu Lai[1,3], Hongxin Ou[1,2] & Peishan Yi [1]✉

The establishment of cell polarity is a prerequisite for many developmental processes. However, how it is achieved during tip growth in plants remains elusive. Here, we show that the RHO OF PLANTs (ROPs), ROP GUANINE NUCLEOTIDE EXCHANGE FACTORs (RopGEFs), and ROP GTPASE-ACTIVATING PROTEINs (RopGAPs) assemble into membrane domains in tip-growing cells of the moss Physcomitrium patens. The confinement of membrane domains requires redundant global inactivation of ROPs by PpRopGAPs and the PLECKSTRIN HOMOLOGY (PH) domain-containing RenGAP PpREN. Unexpectedly, PpRopGAPs and PpREN exert opposing effects on domain size and cell width upon overexpression. Biochemical and functional analyses indicate that PpRopGAPs are recruited to the membrane by active ROPs to restrict domain size through clustering, whereas PpREN rapidly inactivates ROPs and inhibits PpRopGAP-induced clustering. We propose that the activity- and clustering-based domain organization by RopGAPs and RenGAPs is a general mechanism for coordinating polarized cell growth and cell size regulation in plants.

Tip growth is essential for nutrient uptake, fertilization, and signal perception in various cells such as filamentous fungi[1], pollen tubes and root hairs in seed plants[2,3], and protonemata and rhizoids in mosses[4]. The specification of the growing tip is regulated by the conserved Cdc42/Rho/Rac small GTPases, which, in plants, involve the unique Rho-type family RHO OF PLANTS (ROPs)[5]. ROPs polarize tip growth by promoting calcium signaling[6], cytoskeleton remodeling[6,7], vesicle fusion[8], and exocytosis[9]. To date, many ROP effectors have been identified. However, how active ROPs are confined to the growing tip is poorly understood.

Like Cdc42/Rho/Rac GTPases, plant ROPs cycle between an active GTP-bound form and an inactive GDP-bound form[5]. ROP GUANINE NUCLEOTIDE EXCHANGE FACTORS (RopGEFs) and ROP GTPASE-ACTIVATING PROTEINS (RopGAPs) convert ROP-GDP to ROP-GTP and ROP-GTP to ROP-GDP, respectively[10]. In many cells, ROPs are polarly localized and their localization depends on lipid modification,

membrane lipid association, GTP/GDP-binding status, and membrane trafficking[11–15]. Among these, dynamic activation and inactivation by RopGEFs and RopGAPs, respectively, play a central role and have been proposed to involve a self-organization mechanism[13,15]. ROPs have been shown to form membrane clusters in root epidermal cells[16,17], pollen tubes[18], and root hairs[19], suggesting the involvement of self-organization in tip-growing cells.

The self-organization model is based on the fact that ROPs are activated by RopGEFs, and inactivated and restricted by RopGAPs. Although much progress has been made in the regulation of RopGEFs recently[11–15], the roles of RopGAPs in polar domain formation are less understood. There are two structurally distinct subtypes of ROP-related GAPs in land plants[10,20]: one contains a CDC42/RAC-INTER-ACTIVE BINDING (CRIB) motif before the GAP domain and the other bears a PLECKSTRIN HOMOLOGY (PH) domain, a GAP domain, and a coiled-coil (CC) motif-carrying tail, referred to as RopGAP and RenGAP,

[1]Key Laboratory of Bio-Resource and Eco-Environment of Ministry of Education, College of Life Sciences, Sichuan University, No. 24 South Section 1, Yihuan Road, Wuhou District, Chengdu, Sichuan 610064, PR China. [2]Present address: School of Life Sciences, Tsinghua University, Beijing 100084, PR China. [3]These authors contributed equally: Jingtong Ruan, Linyu Lai. ✉e-mail: yipeishan@scu.edu.cn

respectively[5]. RopGAPs and RenGAPs both stimulate GTP hydrolysis by ROPs[21–23], but appear to have differential functions. RopGAPs in Arabidopsis and tobacco localize at the apical membrane of pollen tubes to inactivate ROPs locally[23–25], while Arabidopsis ROP ENHANCER 1 (AtREN1), one of the RenGAPs, globally inhibits ROP signaling in the cytoplasm[22]. A recent study reports that AtREN1 also localizes at the shank membrane of root hairs and interacts with the tip-growth regulator ARMADILLO REPEAT ONLY PROTEINS (AROs)[26,27], implying its potential function as a local inactivator. In Arabidopsis, only two RopGAPs are detectable in pollen tubes. However, their loss of function does not cause discernable growth defects[24]. These findings imply an intricate regulation of ROPs by GAPs and necessitate a comprehensive loss-of-function study of all RenGAPs and RopGAPs. However, the strong male fertility defects in *Atren1* mutants[22] preclude the generation of high-order mutants[26].

The moss *Physcomitrium patens* (*P. patens*) represents one of the extant species closest to the land plant ancestor[28,29]. During the lifecycle of *P. patens*, spores grow into filamentous tissues (protonemata) and develop leafy gametophores at later stages[29]. The protonemata comprise two types of cells termed caulonema cells and chloronema cells, of which the former has fewer and smaller chloroplasts and grows faster[30]. Due to the high regenerative capacity, the protonemata can be indefinitely maintained in the laboratory[31]. This feature, together with the high rate of homologous recombination[32], makes *P. patens* an excellent model for studying tip growth[33]. Previously, ROPs and ROP regulators have been reported to control protonema growth[34–38]. In this study, we show that PpRopGAPs and PpREN redundantly regulate tip growth by globally inhibiting ROP activity. Unexpectedly, they also oppositely regulate cell width. The diverged functions of GAPs suggest that ROP activity and spatial organization are coordinated to establish a polar domain, which may represent a common mechanism for tip growth and cell size regulation in plants.

## Results

### PpRopGAPs localize to the apical membrane and delimit the polar membrane domain

The genome of *P. patens* encodes one RenGAP (PpREN) and six RopGAPs (PpRopGAP1 to PpRopGAP6)[20]. To investigate how PpRopGAPs regulate cell growth, we examined the localization of endogenous PpRopGAPs by fusing the bright green fluorescent protein mNeon-Green (mNG) at the amino- (N-) terminus through homologous recombination. In protonemata, only PpRopGAP1 and PpRopGAP5 were detectable and PpRopGAP1 exhibited a relatively higher expression level. Both proteins were found at the tip of caulonema cells and branch initiation sites (Supplementary Fig. 1a, b). PpRopGAP1 fused with a carboxyl- (C-) terminal Citrine or mNG showed identical localization patterns (Supplementary Fig. 1c–e). Interestingly, PpRopGAP1 signals at the apex were relatively lower than those at the flanking regions (Supplementary Fig. 1d). Upon growth direction change, PpRopGAP1 at one side of the flanking regions extended toward the tip. The actin foci (labeled by Lifeact-mCherry) were disassembled and reassembled at a neighboring cytoplasmic region (Fig. 1a, Supplementary Movie 1). Meanwhile, the redistributed PpRopGAP1 generated a new low-intensity zone at the tip. During branch initiation and growth, a low-intensity zone was not observed at the expanding apex, although the fluctuation of actin foci was observed as seen in tip cells (Supplementary Fig. 1d, f).

As PpROPs and PpRopGEF4 are enriched at the apical membrane[35,36,39], we labeled the endogenous PpRopGEF4 and PpROP4 with a C-terminal mCherry (PpRopGEF4-mCherry) and an internal mNG (PpROP4-mNG), respectively, and compared their localization patterns with PpRopGAP1. As shown in Fig. 1b–d, PpRopGEF4 was exclusively found at the apical dome. PpROP4 displayed a similar enrichment but labeled a slightly broader region and showed weak fluorescence on the remaining membrane. Noticeably, PpRopGAP1 marked a much wider region than PpRopGEF4. Similar to PpRopGAP1,

PpRopGEF4 and PpROP4 displayed a small low-intensity zone at the apex. Time-lapse imaging revealed signal oscillation at the low-intensity zone (Supplementary Movies 2 and 3), implying that the apex is an actively remodeling region. On the membrane, all proteins were present in mobile particles of nanoscale size (< 1 μm) (Fig. 2a) and the PpRopGAP1 particles were more visible than those formed by PpROP4 or PpRopGEF4 (Supplementary Movie 4). In addition, 79% (n = 42 particles from 9 cells) and 80% (n = 44 particles from 9 cells) of PpROP4 and PpRopGAP1 particles, respectively, exhibited colocalization with PpRopGEF4 particles (Fig. 2g), implying that they are largely present in the same clusters. Taken together, these data indicate that endogenous ROPs, RopGEFs, and RopGAPs are organized into membrane domains with the boundary marked by RopGAPs. They rapidly diffuse and potentially form protein clusters within the domain.

### PpRopGAP1 is recruited to transient membrane domains following PpRopGEF4 and PpROP4

To investigate the ability of PpROP4, PpRopGEF4, and PpRopGAP1 in forming domains, we next investigated their spatiotemporal relationship during polarity initiation in pre-branching subapical cells. As shown in Supplementary Fig. 1g, h, all proteins labeled the future branching site before bulge initiation. As in tip cells, PpRopGAP1 marked a broader region than PpRopGEF4. Interestingly, using time-lapse imaging we detected ectopic transient domains marked by PpROP4, PpRopGEF4, and PpRopGAP1 (Fig. 2b–f). These domains formed either at the apicolateral membrane of the subapical cell or at the basolateral membrane of the tip cell and persisted for several to tens of minutes. In some cases, the transient domain moved along the membrane and underwent disassembly and reassembly, suggesting that they were unstable. Nevertheless, PpROP4 and PpRopGEF4 were almost simultaneously recruited to these domains with PpROP4 occurring in a slightly wider time window (Fig. 2b, c, Supplementary Movie 5). By contrast, PpRopGAP1 was recruited much later than PpRopGEF4 (Fig. 2d–f, Supplementary Movie 6). These observations suggest that ROP-mediated domain assembly is initiated by unstable clustering of ROPs and RopGEFs and is presumably reinforced by subsequently recruited RopGAPs.

### The N-terminal CRIB motif and a conserved pre-CRIB motif are necessary and sufficient for the membrane targeting of PpRopGAP1

We next investigated how PpRopGAP1 was targeted to membrane domains. Plant RopGAPs comprise a CRIB motif-containing amino terminus (Nter) and a GAP domain-containing carboxyl terminus (Cter)[10,20]. When tagged with mNG, the Nter portion of PpRopGAP1 (amino acid 1-193) was localized to the apical membrane and displayed an enrichment similar to the full-length PpRopGAP1 (Fig. 3a). Introducing a histidine-to-alanine mutation in the core CRIB motif (H165A)[21] or removing the Nter portion (Δ1-193) completely abolished membrane localization, suggesting that polar localization of PpRopGAP1 is primarily mediated by the CRIB domain. Interestingly, when GAP activity was inhibited by an arginine-to-leucine mutation (R239L)[23], the membrane localization of PpRopGAP1 was also lost (Fig. 3a). It is plausible that the mutated GAP domain may adopt a conformational change that hinders the binding of the CRIB domain with ROPs. As CRIB domains preferentially bind active ROPs[21,23], these results suggest that RopGAPs are recruited to the membrane by active ROPs.

We next investigated the minimal portion of the CRIB domain for membrane targeting. The CRIB motif has a 16-amino-acid core peptide in animals[40] and is present in plant RICs and RopGAPs[21,23,41]. The fully functional CRIB domain in plants appears to require additional sequences[23,41,42], which have not been unequivocally identified. To characterize the functional region of a CRIB domain in RopGAPs, we obtained protein sequences carrying a CRIB motif and a RhoGAP domain from the InterPro database and performed alignment analysis.

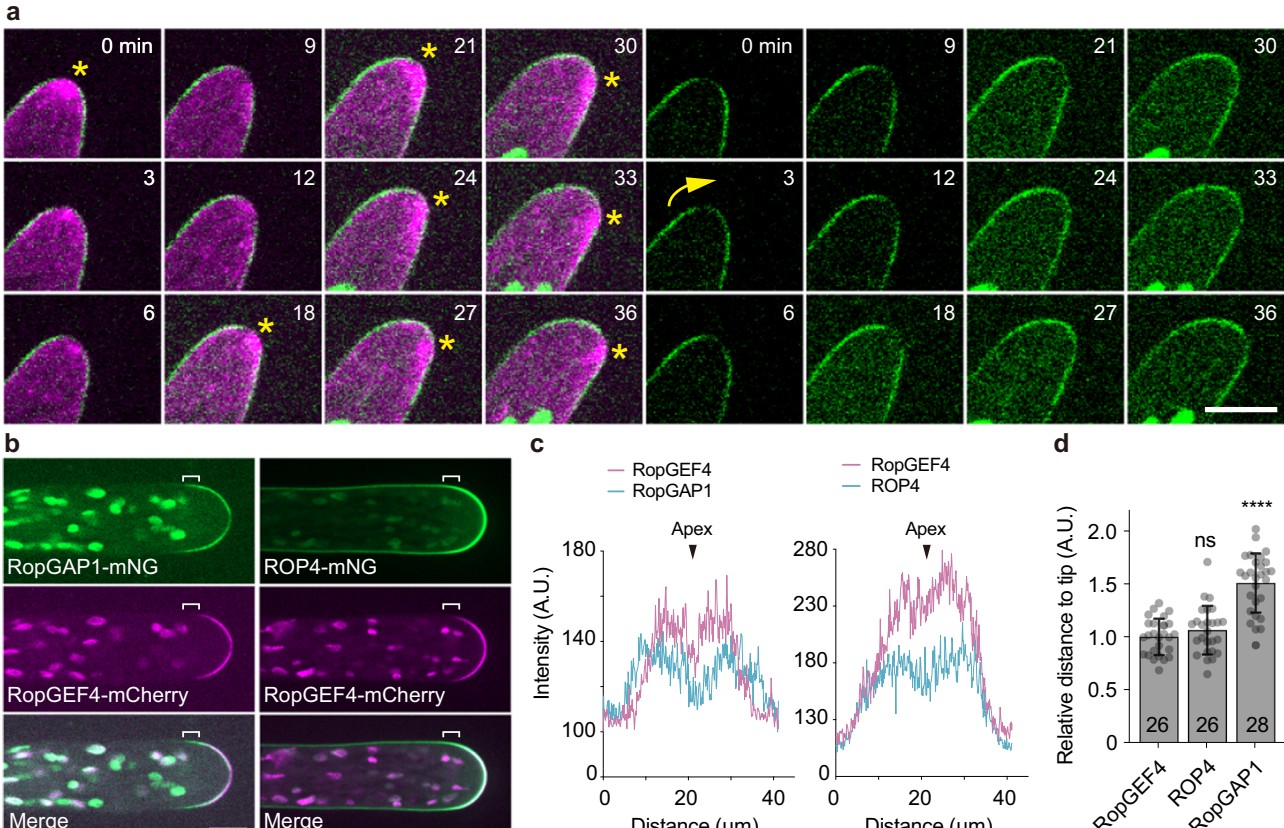

**Fig. 1 | The endogenous PpRopGAP1 is enriched at the tip of caulonema cells and delimitates the dome region marked by PpROP4 and PpRopGEF4. a** Time-lapse images of PpRopGAP1 fused with the bright green fluorescent protein mNeonGreen (mNG, green) and actin labeled with Lifeact-mCherry (magenta). Note that PpRopGAP1 (green) exhibits a low-intensity area at the apex near the cytoplasmic actin foci (star) and reorients its position (arrow) when the growth direction changes. **b** Colocalization of PpRopGAP1, PpRopGEF4, and PpROP4 in tip cells. The endogenous PpRopGEF4 and PpROP4 are fused with a C-terminal mCherry and an internal mNG, respectively. PpRopGEF4 is exclusively localized at the apical dome. PpROP4 is enriched around the same place but also weakly labels other membrane regions. PpRopGAP1 occupies a broader area than PpRopGEF4 toward the base. The flanking regions of the PpRopGEF4 signal are indicated with white brackets. **c** Intensity plots of PpRopGAP1, PpRopGEF4, and PpROP4 along the apical membrane. Note that PpRopGEF4 and PpROP4 also exhibit a low-intensity area at the apex. Scale bars:10 μm in all panels. **d** Quantification of the relative distance of enriched PpROP4 and PpRopGAP1 signals to the cell tip compared with PpRopGEF4. The number of cells for quantification is shown on the bars. Data are presented as mean values ± SD with individual points shown. Statistical analyses were performed using adjusted one-way ANOVA tests. ns, not significant. ****$p < 0.0001$. The exact p-values are available in Source Data. Source data are provided as a Source Data file.

RopGAPs are present in land plants (Embryophyta), green algae *Klebsormidium nitens*, and protists of the Sar supergroup (stramenopiles, alveolates, and rhizaria) (Supplementary Data 1, Supplementary Fig. 2). In addition to the CRIB motif, we revealed a highly conserved region comprising ~20 amino acids before the CRIB motif (hereafter referred to as the PCRIB motif). The PCRIB motif contains abundant hydrophobic residues and a conserved (R/K)(R/K)SX₄C module (Fig. 3b) and is predicted to adopt an α-helix fold (Supplementary Fig. 3)[43]. Interestingly, RICs also contain a conserved region before the CRIB motif (Supplementary Fig. 4a)[42]. Although the corresponding sequences are not similar to the PCRIB motifs in RopGAPs, they also tend to form α-helical structures (Supplementary Fig. 4b).

We mutated the two conserved arginine residues (R134 and R135) to leucine and assessed their effects on PpRopGAP1-PpROP4 binding in yeast-two-hybrid assays. Although R134L or R135L alone did not abolish the interactions between PpRopGAP1 and constitutively active PpROP4 (ROP4CA), the interaction was lost when both mutations were introduced (Fig. 3c). Truncated forms of the CRIB domain containing either the CRIB motif alone (S150-S193) or the PCRIB plus CRIB motif (Q122-S193) showed interactions with PpROP4CA. However, the interaction between the CRIB motif-only fragment and PpROP4 was relatively weak (Fig. 3c). When the H165A mutation was introduced, the binding ability of the CRIB motif was completely blocked. As the PCRIB

motif is rich in hydrophobic residues, we speculate that the R134/135 L mutation may disrupt intramolecular interactions and influence protein stability. Indeed, the R134/135 L double-mutant protein was poorly expressed in contrast to those carrying one mutation (Fig. 3d), although the expression at the transcription level was not affected (Supplementary Fig. 5). Moreover, we failed to express R134/135L mutant proteins using bacterial expression systems in vitro. These data suggest that the PCRIB motif plays a critical role in protein folding. We next expressed the mutated forms of the CRIB domain in planta and examined their localization. Interestingly, the CRIB motif alone was only found in the cytoplasm (Fig. 3e). The addition of the PCRIB motif led to a membrane localization pattern similar to wild-type (WT) PpRopGAP1, albeit the overall expression was lower. Importantly, the R134L or R135L mutation severely impaired membrane localization, indicating that the PCRIB motif is part of a functional CRIB domain in vivo. Based on these results, we concluded that the membrane targeting of PpRopGAP1 depends on the CRIB core motif and is facilitated by a conserved PCRIB motif that stabilizes protein conformation.

## PpRopGAPs and PpREN redundantly regulate tip growth and development

To investigate how PpRopGAPs regulate tip growth, we generated mutants of all PpRopGAPs by introducing mutations in the CRIB motif

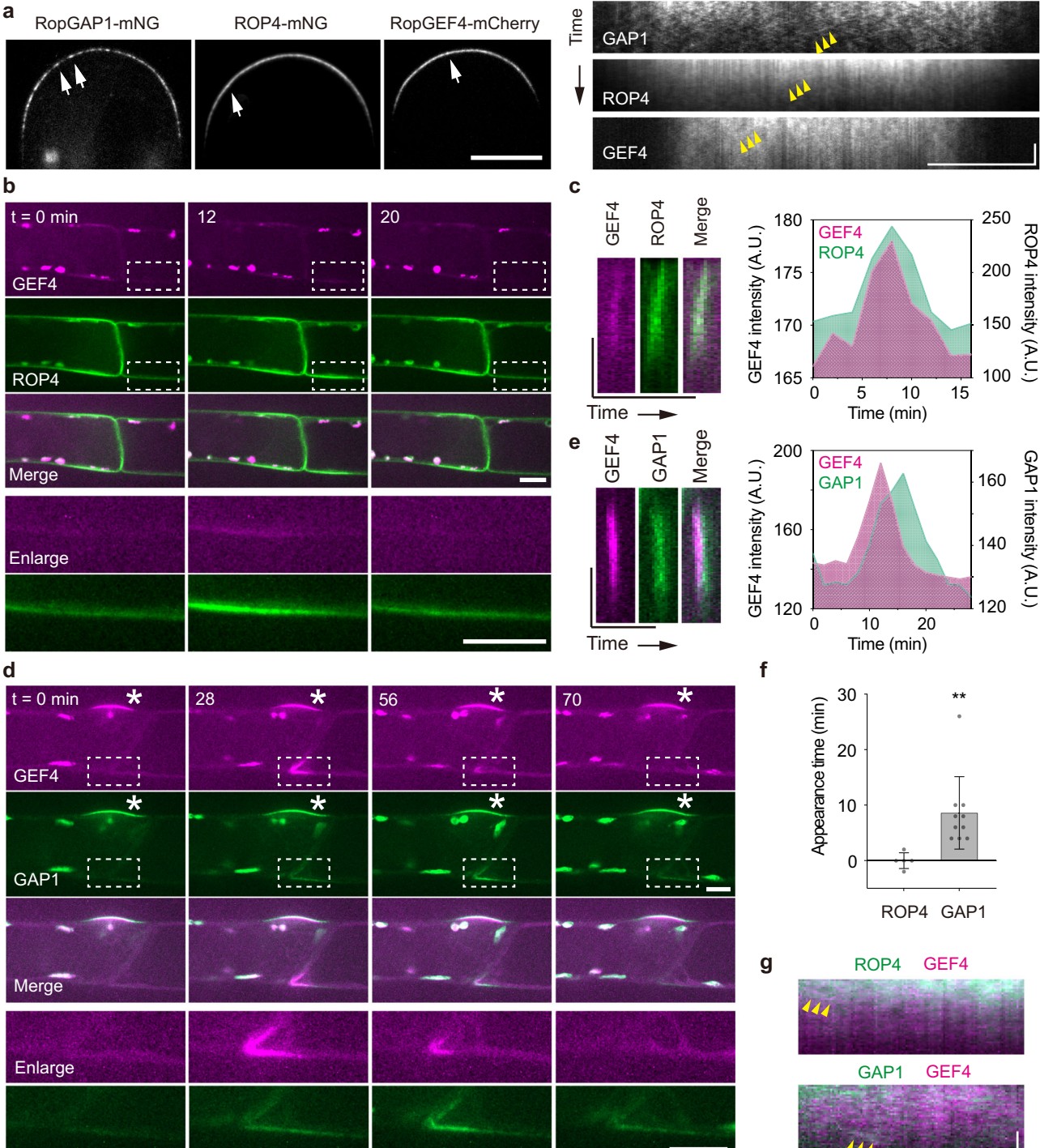

**Fig. 2 | PpROP4, PpRopGEF4, and PpRopGAP1 are sequentially recruited into membrane domains. a** PpRopGAP1, PpROP4, and PpRopGEF4 are present in mobile particles (white arrows) on the apical membrane. Yellow arrowheads indicate the movement of representative particles. Note that PpRopGAP1 particles are more visible. Scale bars: horizontal, 10 μm; vertical, 10 s. **b** Transient clustering of PpROP4 and PpRopGEF4. PpROP4 is enriched at the lateral surfaces of subapical cells (left) before branching. A transient accumulation (dashed rectangles, enlarged at the bottom panel) occurs at the basal membrane of the tip cell. Scale bar: 10 μm. **c** Kymographs showing PpROP4 and PpRopGEF4 recruitment at the transient membrane domain. Averaged intensities of PpROP4 and PpRopGEF4 in kymographs over time are plotted. Scale bars: vertical, 10 μm; horizontal, 1 h. **d** Transient clustering of PpRopGAP1 and PpRopGEF4. PpRopGAP1 and PpRopGEF4 are localized at the branch initiation site of subapical cells (stars). A transient accumulation

(dashed rectangles, enlarged at the bottom panel) occurs at the basal membrane of the tip cell. Scale bar: 10 μm. **e** Kymographs showing PpRopGAP1 and PpRopGEF4 recruitment at the transient membrane domain. Note that PpRopGEF4 appears earlier than PpRopGAP1. Scale bars: vertical, 10 μm; horizontal, 1 h. **f** Quantification of PpROP4 ($n = 5$) and PpRopGAP1 ($n = 10$) appearance time relative to PpRopGEF4. Data are presented as mean values ± SD with individual points shown. Statistical analysis was performed using a two-tailed student's t-test. **$p < 0.01$. The exact $p$-values are available in Source Data. **g** Colocalization of PpROP4 and PpRopGAP1 with PpRopGEF4 in mobile particles. Two representative particles are indicated by arrowheads in the kymographs. Scale bars: horizontal, 1 μm; vertical 10 s. Similar results in (**a, b, d**) were obtained in at least three independent experiments. Source data are provided as a Source Data file.

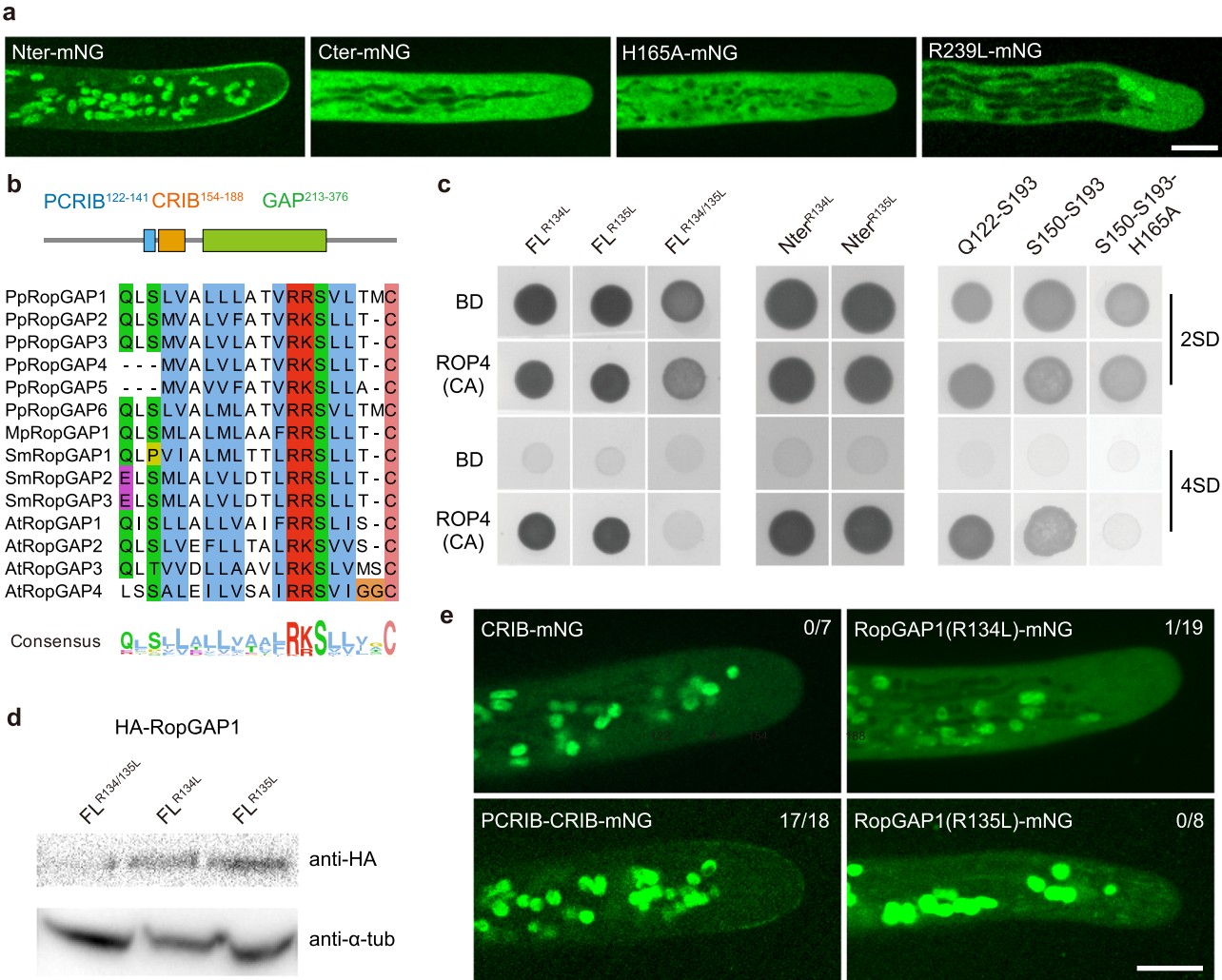

**Fig. 3 | The membrane recruitment of PpRopGAP1 requires its CRIB motif, GAP activity, and a pre-CRIB motif. a** Localization of the CRIB domain-containing N-terminal portion (Nter, 1-193), the GAP domain-containing C-terminal portion (Cter, 194-469), and full-length PpRopGAP1 carrying a histidine-to-alanine (H165A) or an arginine-to-leucine (R239L) mutation. Scale bar: 10 μm. **b** The conserved pre-CRIB (PCRIB) motif identified in RopGAPs. The consensus sequence is generated from 1 063 RopGAPs. Pp, *Physcomitrium patens*; Mp, *Marchantia polymorpha*; Sm, *Selaginella moellendorffii*; At, *Arabidopsis thaliana*. The PCRIB motif is not present in AtRop-GAP5. **c** The PCRIB motif facilitates the interaction between the full-length (FL) PpRopGAP1 and constitutively active PpROP4 (ROP4CA) in yeast-two-hybrid assays. Mutating two conserved residues (R134L and R135L) blocks PpRopGAP1-PpROP4CA interaction. Note that the absence of the PCRIB motif weakens but does not abolish the binding between PpROP4CA and the Nter portion of PpRopGAP1. **d** PpRopGAP1(R134/135L) exhibits reduced protein stability. The PpRopGAP1 tagged with hemagglutinin (HA) was detected by western blotting. Tubulin was blotted as a loading control. **e** Localization of the CRIB motif (S150-S193), the PCRIB plus CRIB motif (Q122-S193), and PpRopGAP1 carrying the R134L or R135L mutation. Note that only the PCRIB plus CRIB motif exhibits a polar membrane localization, albeit its expression level is relatively low. The number of cells with a membrane localization pattern is shown. Scale bar: 10 μm. Similar results in (**a**, **d**, **e**) were obtained in at least three independent experiments. Source data are provided as a Source Data file.

or the GAP domain using the clustered regularly interspaced short palindromic repeats (CRISPR)/Cas9-based genome editing technology[44]. We first obtained a quintuple mutant carrying frameshift mutations in PpRopGAP1/3/6 and in-frame deletions in PpRopGAP2/4. This quintuple mutant appeared superficially wild-type, suggesting a functional redundancy between PpRopGAPs. Further editing of PpRopGAP2/4/5 with oligonucleotide templates[45] allowed us to obtain two types of sextuple mutants: one carrying frameshift mutations in all PpRopGAPs (hereafter referred to as *ropgap* mutant), and the other containing in-frame deletions in PpRopGAP2/4 but frameshift mutations in other PpRopGAPs (referred to as *ropgap hypomorphic mutant* or *ropgap-HM*) (Supplementary Fig. 6). The presence of frameshift mutations suggests that the *ropgap* mutant is potentially null. To verify this, we inserted an mNG or a Citrine tag to the C-terminus of endogenously mutated PpRopGAP1. As expected, we did not detect any expression (Supplementary Fig. 7a, b). Furthermore, overexpression of

the mutated PpRopGAP1 fused with mNG did not produce detectable fluorescence either (Supplementary Fig. 7c). These results strongly indicate that PpRopGAP expression is substantially blocked at the protein level by frameshift mutations and the *ropgap* mutant is null. Interestingly, the growth of protonemata and gametophores was only marginally affected in the *ropgap* mutant (Fig. 4a; Supplementary Fig. 8a). Therefore, PpRopGAPs are not essential for *P. patens* viability and may play a minor role in development.

Because the knockout of PpREN in *P. patens* (hereafter referred to as *ren* mutant) via homologous recombination revealed no discernable growth defects (Fig. 4a), we suspected that PpREN and PpRopGAPs have redundant roles, and thus knocked out *ren* in the *ropgap* or *ropgap-HM* background. As expected, the *ropgap, ren* and *ropgap-HM, ren* septuple mutants exhibited strong defects in protonema growth and gametophore development when cultured on standard BCDAT medium (Fig. 4a–f; Supplementary Fig. 8a–c). The protonema growth

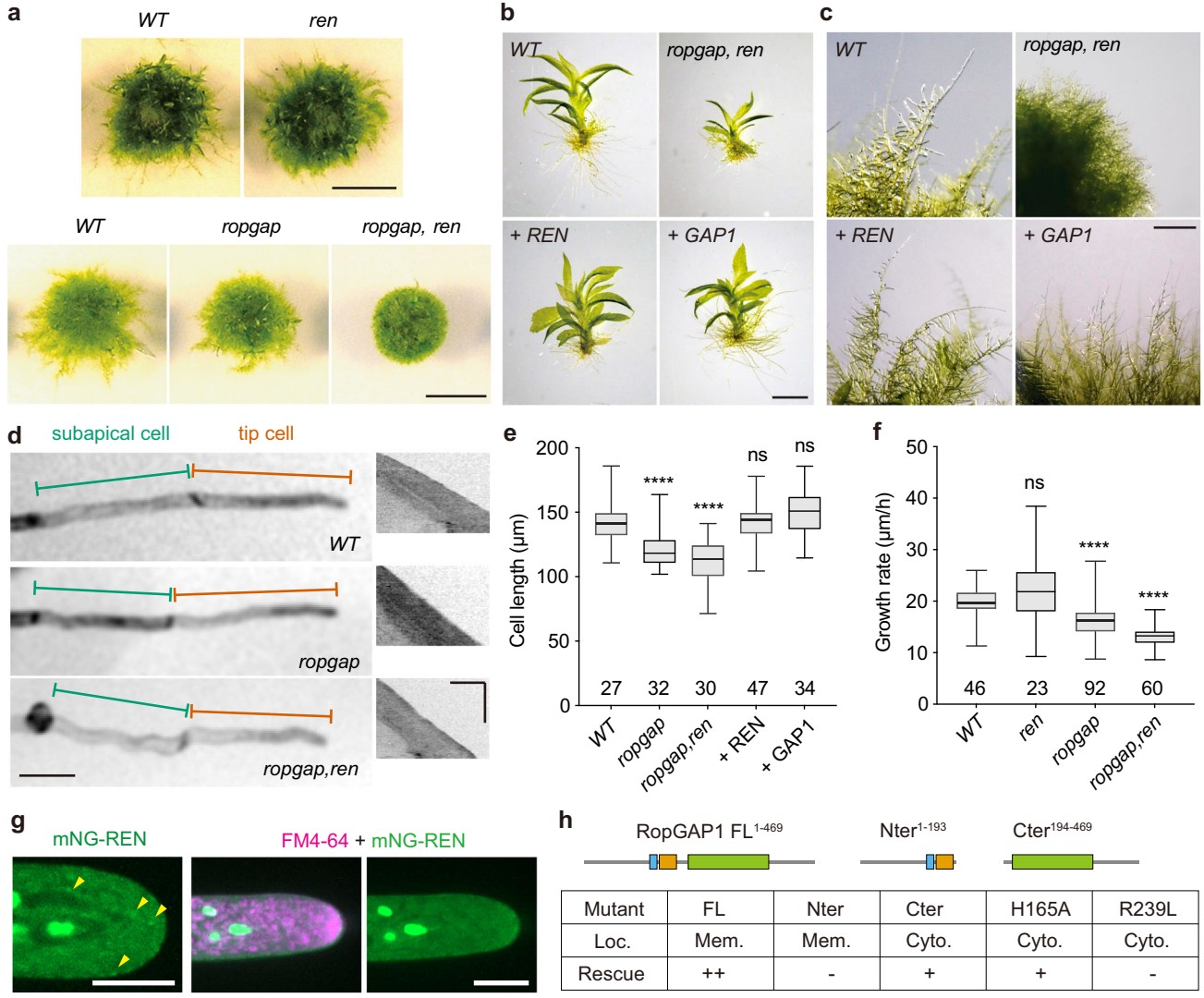

**Fig. 4 | PpRopGAPs and PpREN redundantly regulate tip growth and development. a** Colony growth of wild-type (WT) and mutant mosses. Scale bars: 0.5 cm. **b** Gametophores of WT, mutant, and rescued mosses. Cerulean-fusion PpREN or PpRopGAP1 (simplified as + REN and + GAP1) were expressed for rescue in the *ropgap, ren* septuple mutant background. Scar bar: 1 mm. **c** Cell morphology and growth of WT, mutant, and rescued mosses. Scale bar: 1 mm. **d** Tip cell growth in WT and mutant mosses. Representative kymographs of growing tip cells are shown (right). Note that cells in the *ropgap, ren* mutant are wavy. Scale bars: horizontal, 50 μm; vertical, 5 h. **e** Quantification of cell length in WT, mutant, and rescued mosses. **f** Quantification of growth rate in WT and mutant tip cells. **g** Localization of PpREN in the cytoplasm. Note that PpREN occasionally forms puncta-like structures but does not exhibit obvious enrichment or colocalization with FM4-64-labeled vesicles at the tip. Scale bars: 10 μm. **h** Summary of localization and rescue capability of full-length (FL) and mutant forms of PpRopGAP1. Note that PpRopGAP1

can partially rescue cell growth phenotypes in the absence of the Nter portion or a fully functional CRIB domain. The rescue capability was categorized as either complete (++), partial (+), or absent (−) when mosses exhibited fully, partially, or no elongated caulonema cells on BCDAT plates, respectively. Protein localization was determined by assessing the relative intensity on the membrane (Mem.) and in the cytosol (Cyto.). If the membrane signals were visually stronger than the cytosolic fluorescence, they were classified as membrane localization. Data are presented as box-and-whisker plots, showing the interquartile range (box), the median (horizontal line), and minimum and maximum values (whiskers) in (**e**, **f**). The number of cells used for quantification in (**e**, **f**) is shown at the bottom. Statistical analyses were performed using one-way ANOVA tests. ns, not significant. ****$p < 0.0001$. The exact *p*-values are available in Source Data. Similar results in (**g**) were obtained in three independent experiments. Source data are provided as a Source Data file.

defects were due to a significant loss of fast-growing caulonema cells (Fig. 4c) and could be rescued by overexpressing PpREN or PpRopGAP1 fused with an N-terminal Cerulean, demonstrating the functional redundancy between PpREN and PpRopGAPs (Fig. 4b, c, e; Supplementary Fig. 8d). When cultured on the nutrient-limiting BCD medium, which stimulates caulonema cell development, mutant plants developed wavy filaments. The caulonema cells were shorter and grew slower than WT cells (Fig. 4d–f). Similarly, the *ropgap-HM, ren* mutants exhibited a decrease in cell length and growth and could not regenerate efficiently after protoplasting (Supplementary Fig. 8e–g). These findings indicate that PpREN and PpRopGAPs play overlapping roles in regulating tip growth and development in *P. patens*.

We have previously shown that overexpressed PpREN is mostly localized in the cytoplasm[36]. This fact contradicts the functional redundancy between PpREN and PpRopGAPs. We examined the endogenous PpREN with mNG fusion and confirmed its cytoplasmic localization (Fig. 4g). Although we occasionally observed dot-like structures in the cytoplasm or associated with membranes, we did not detect clear enrichment or colocalization of PpREN1 with vesicles, as has been reported for the Arabidopsis AtREN1[22,26]. We next examined the function of membrane localization of PpRopGAP1 by performing rescue experiments. The Nter portion and the inactive form of PpRopGAP1 did not rescue cell growth phenotypes (Fig. 4h). Interestingly, PpRopGAP1 mutants carrying the H165A mutation or lacking

the Nter portion could partially rescue growth phenotypes although they completely lost membrane localization (Fig. 3a). These results indicate that RopGAPs can globally inactivate ROPs as RenGAPs do. More importantly, local inactivation of ROPs by RopGAPs is not essential for polarity formation and tip growth (Fig. 4h).

## PpRopGAPs and PpREN restrict branch initiation

The wavy filaments in *ropgap, ren* mutants imply defects in cell morphology regulation. We thus examined the development of side branches. In WT cells, side branches sequentially develop on subapical cells with the latest one formed commonly on the second subapical cell. We quantified the percentages of subapical cells that had developed a side branch. In the *ropgap* sextuple mutant or *ren* mutant alone, branch formation was normal. However, there was a clear increase of branches on the second subapical cell in *ropgap, ren* septuple mutants (Fig. 5a). Complementation with PpREN fully rescued the branching phenotype, while the overexpression of PpRopGAP1 drastically inhibited branch formation (Fig. 5a). PpRopGAP1 overexpression in WT plants induced a similar effect, demonstrating that excess PpRopGAP1 can inhibit branching. These data indicate that branch initiation requires an optimal level of active ROPs and PpRopGAP1 has a stronger effect than PpREN in limiting branch formation. This notion is consistent with the localization of PpRopGAPs rather than PpREN at the branch initiation sites (Fig. 2d; Supplementary Fig. 1), where they may locally inactivate and cluster ROPs.

In addition to branch formation phenotypes, we found an increase in bulge width in *ropgap, ren* mutants (Fig. 5b, c). Since bulge size can affect the positioning of the division plane[36,46], we performed time-lapse imaging to examine the division process. Interestingly, nuclear migration and cell division were largely normal. However, during cytokinesis, the phragmoplast frequently attached to one side of the basal region of bulges and expanded directionally toward the other side (Fig. 5d, Supplementary Movie 7). Although cell division was completed, cytokinesis was significantly delayed due to an increase in the duration of phragmoplast expansion (Fig. 5e). We concluded that PpREN and PpRopGAPs are not essential for phragmoplast guidance but could indirectly influence cell division by altering bulge morphology. In support of this notion, PpRopGAP1 overexpression strongly delayed bulge emergence (Fig. 5a) and could lead to division plane mispositioning when the cells failed to initiate bulges (Fig. 5f). Such phenotypes are also observed in *rop* mutants or cells treated with actin inhibitors[36,46], supporting that PpRopGAPs, PpROPs, and actin may function in a common pathway. Interestingly, the width and height of bulges in PpRopGAP1 overexpression lines were not significantly different from those of WT cells (Fig. 5c). Therefore branching defects are likely caused by reduced polarity initiation rather than by global inhibition of polar cell growth. This notion is in line with the facts that PpRopGAP1 overexpression does not markedly affect cell length (Fig. 4f) and the apical growth of bulges (i.e. height in Fig. 5c) is normal in the *ropgap, ren* septuple mutant. Taken together, these results indicate that PpREN and PpRopGAPs function to restrict branch initiation and suggest that polarity initiation and polar cell growth have different sensitivities to ROP activity, with the former more sensitive to ROP inactivation and the latter more sensitive to ROP hyperactivation.

## Membrane distribution of PpROP4 is altered in the ropgap, ren mutants

The overall phenotypes in *ropgap, ren* mutants suggest a dysregulation of cell polarity. Therefore we examined the localization of PpROP4 in tip cells. Surprisingly, the polar localization of PpROP4 was only mildly affected (Fig. 6a). The boundary between the apical dome and basal membrane became less evident (Fig. 6b). Moreover, we observed a decrease in PpROP4 enrichment at the tip (Fig. 6c). These results suggest that ROPs are less concentrated at the apical region in the absence of PpRopGAPs and PpREN. Interestingly, before tip cell

division, we observed early accumulation of PpROP4 in *ropgap, ren* mutants (11/11 cells) at the lateral membrane around the interphase nucleus (Fig. 6d). If the cells were bent, stronger accumulation was always found at the protruding side. After cytokinesis, a polar localization was established at the apical end of the basal daughter cell (i.e. subapical cell), reminiscent of the branch initiation site. Such a phenomenon was never observed in WT cells. These data are consistent with the premature emergence of side branches in *ropgap, ren* mutants (Fig. 5a). Taken together, our results indicate that PpRopGAPs and PpREN facilitate ROP concentration at the polarizing domain but inhibit domain initiation.

## RopGAPs and REN play antagonistic roles in cell width regulation

Although loss-of-function analyses reveal redundant roles of PpRopGAPs and PpREN in tip growth and branch formation, the dramatic effects of ectopically expressed PpRopGAP1 rather than PpREN on branch reduction suggest that their functions are not identical (Fig. 5a, f). Indeed, when PpREN or PpRopGAP1 was overexpressed in *ropgap, ren* septuple mutants, they fully rescued cell growth phenotypes (Fig. 4c, e) but oppositely increased and decreased cell width, respectively (Fig. 7a, b). To verify such a striking difference, we overexpressed PpREN and PpRopGAP1 in WT plants and confirmed that they could oppositely influence cell width (Fig. 7c). Interestingly, we did not detect a significant change in cell width in the *ren* or *ropgap* mutant alone. Cell width was only marginally increased in *ropgap, ren* septuple mutants (Fig. 7b). Because loss-of-function and gain-of-function of ROPs can both lead to short and round cells[34–37,39], we reasoned that cell width regulation requires an optimal amount of active ROPs and this process exhibits distinct sensitivities to changes in the levels of active or inactive ROPs. The increase of active ROPs in *ropgap* or *ren* mutants was not sufficient to induce a change in cell width, while this effect could be enhanced in *ropgap, ren* septuple mutants likely due to their redundant functions in globally inactivating ROPs. By contrast, the inactivation of ROPs by excess PpREN or PpRopGAP1 could effectively alter cell width and the opposite effects are due to their distinct binding capacities and GAP activities (see below and Discussion).

## PpRopGAP1 exhibits higher binding capacity to active PpROP4 than PpREN

To explore how PpRopGAP1 and PpREN interact with ROPs, we performed yeast-two-hybrid analyses and pull-down experiments. As PpROPs are functionally redundant[20,34,36], we only focused on PpROP4. As expected, the full-length PpRopGAP1 interacted with PpROP4CA (represented by a G15V mutation), but not the WT or dominant-negative form (PpROP4DN, represented by a D20N mutation) in yeast-two-hybrid assays[47,48] (Fig. 8a). The Nter portion (1-193) and the Cter portion (194-469) showed similar binding ability and specificity (Fig. 8b). Due to this dual-binding capacity, the H165A or R239L mutation of PpRopGAP1 alone did not abolish the interaction. However, when R239L was introduced into the Cter, the binding with PpROP4CA was abolished. In the pull-down experiments, PpRopGAP1 fused with maltose binding protein (MBP) displayed preferential interactions with Glutathione S-transferase (GST)-tagged PpROP4CA and only weakly interacted with PpROP4 or PpROP4DN (Fig. 8c, Supplementary Fig. 9a). The H165A or R239L mutants exhibited a strong and a mild decrease in binding capacity, respectively (Fig. 8d). Interestingly, we failed to detect interactions between PpROP4CA and PpREN or between PpROP4CA and the PpREN GAP domain in yeast-two-hybrid analyses (Supplementary Fig. 9b). PpROP4CA-PpREN interactions could only be detected via western blotting after pull-down (Fig. 8e). These data suggest that PpREN has a much weaker binding ability than PpRopGAP1 toward active PpROP4. However, our western blotting analyses did not reveal differences in the binding

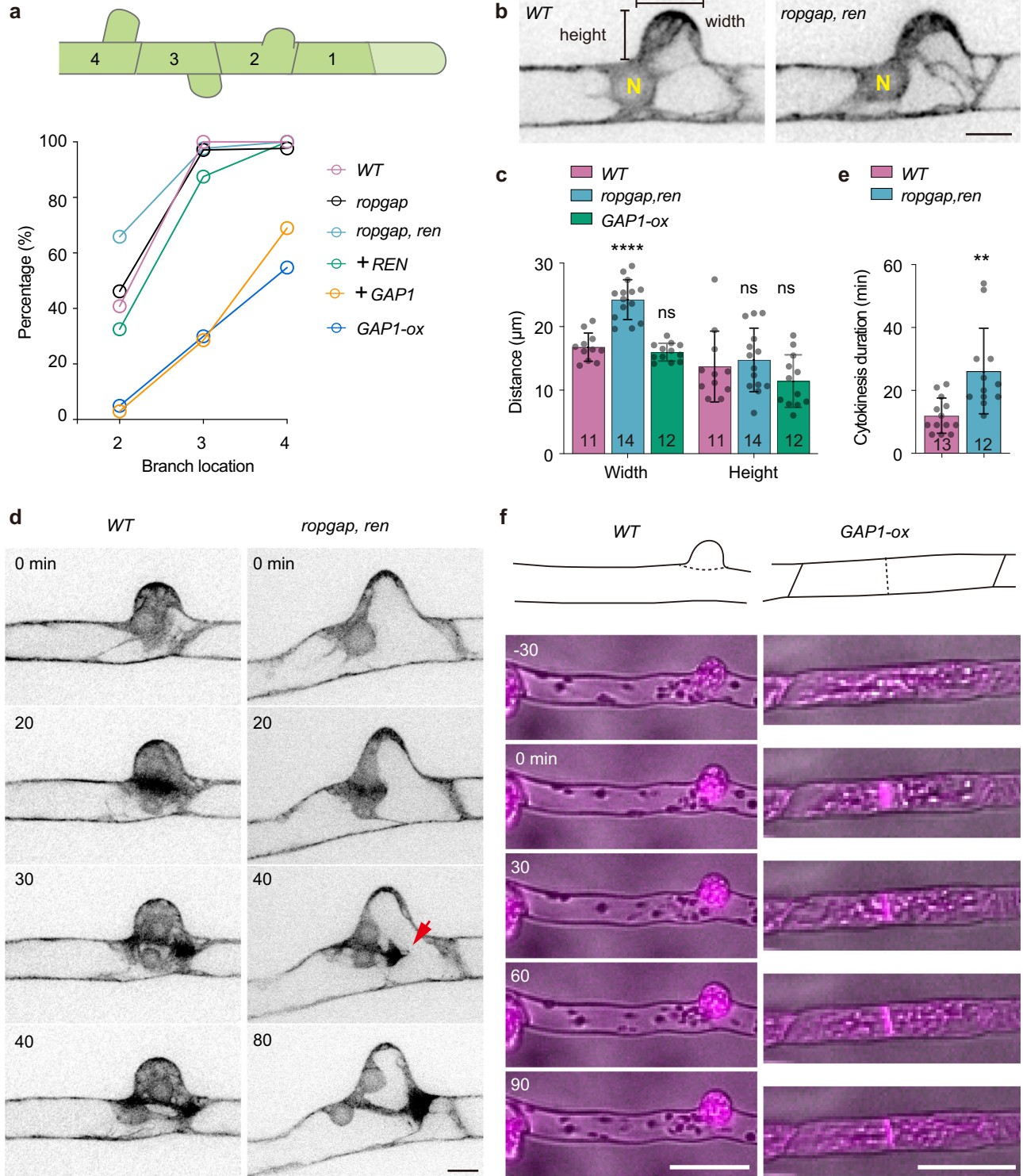

**Fig. 5 | PpRopGAPs and PpREN inhibit branch initiation. a** Percentages of subapical cells that have developed side-branches in WT, mutant, and rescued mosses. Subapical cells are numbered according to their relative position to the tip cell. The number of cells used for quantification ranges between 12 and 80. Note that early formation of branches on the second subapical cells occurs in *ropgap, ren* mutants which could be rescued by PpREN. PpRopGAP1 overexpression not only inhibits premature branching but also strongly blocks branch formation in *ropgap, ren* mutants and WT cells. **b** Morphology of the bulges in WT and mutant subapical cells. N, nucleus. Scale bar: 10 μm. **c** Quantification of bulge width and height. **d** Mitotic division of WT and mutant subapical cells. Cells are visualized with Lifeact-mCherry. The phragmoplast leading edge is indicated (red arrow). Scale bar: 10 μm. **e** Quantification of phragmoplast expansion time. **f** Representative subapical cell divisions in the WT and PpRopGAP1 overexpression lines. Cells are labeled by Lifeact-mCherry and imaged with epifluorescence and bright light. Scale bars: 50 μm. Data are presented as mean values ± SD with individual points shown in (**c**, **e**). The number of cells used for quantification is shown at the bottom. Mutant or overexpression lines were statistically compared with WT using two-tailed student's t-tests. ns, not significant. ****$p < 0.0001$. **$p < 0.01$. The exact p-values are available in Source Data. Similar results in (**f**) were obtained in three independent experiments. Source data are provided as a Source Data file.

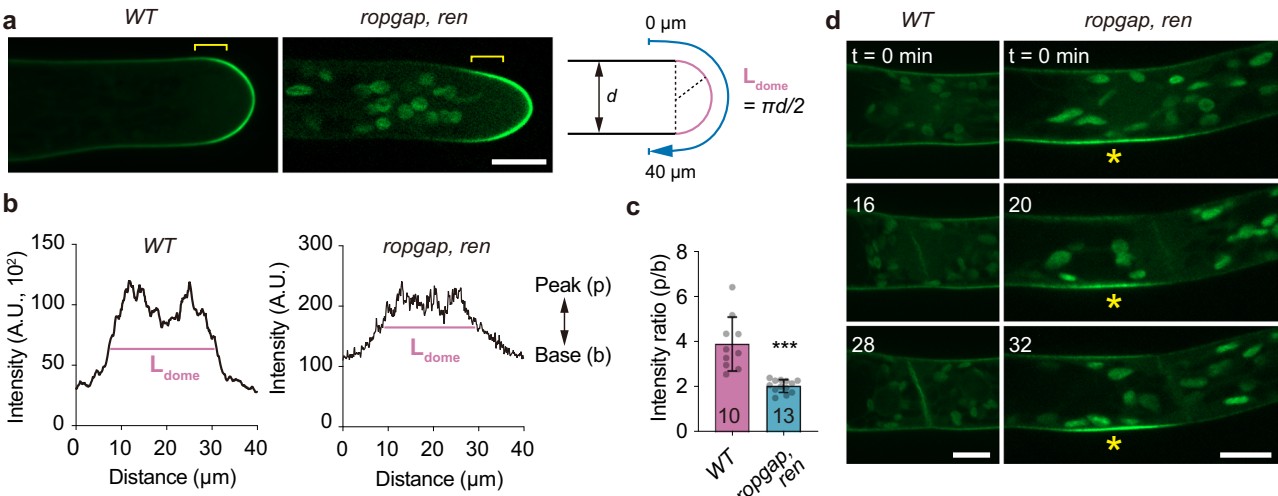

**Fig. 6 | PpROP4 localization is altered in *ropgap, ren* mutants. a** Localization of PpROP4-mNG in WT and *ropgap, ren* mutant tip cells. Note that the boundary (brackets) between the apical dome and subapical membrane is less obvious in mutant cells. Scale bar: 10 μm. **b** Representative intensity plots of PpROP4 along the apical membrane. Intensities were measured along the blue lines shown in (**a**). The dome region is marked by pink lines. **c** Quantification of the intensity ratio of peak and base in the plots. The number of cells used for quantification is shown at the bottom. Data are presented as mean values ± SD with individual points shown. The comparison was performed using a two-tailed student's t-test. ***p < 0.001. The exact *p*-values are available in Source Data. **d** Early accumulation of PpROP4 at branching sites in subapical cells (indicated by stars) of *ropgap, ren* mutants. Images were taken starting from the nuclear envelope breakdown. Scale bar: 10 μm. Similar results in (**d**) were obtained in three independent experiments. Source data are provided as a Source Data file.

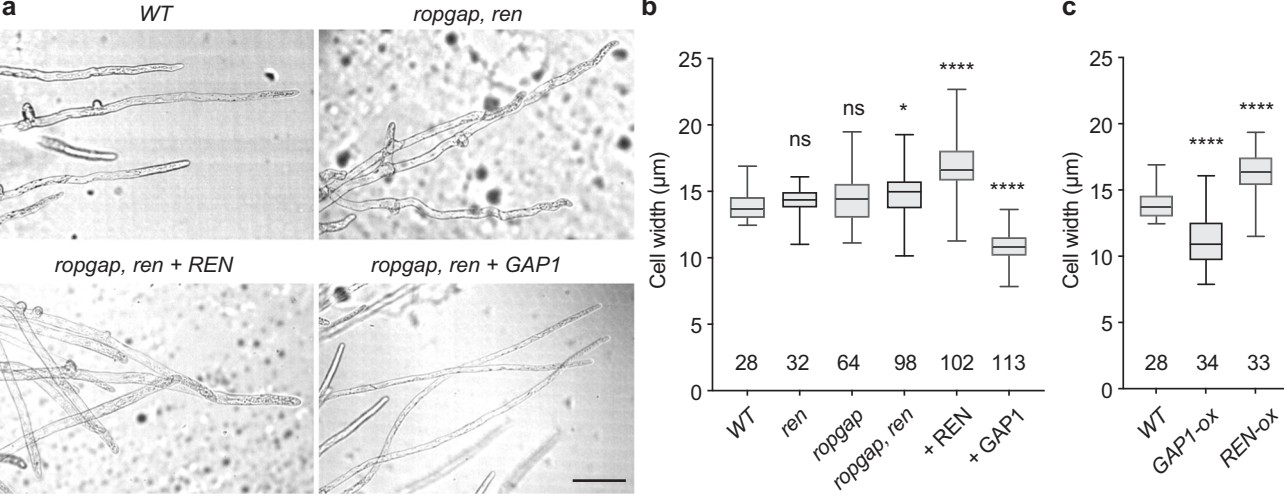

**Fig. 7 | PpRopGAPs and PpREN oppositely regulate cell width. a** Cell morphology and branch pattern in WT, mutant, and rescued mosses. Note that the overexpression of PpREN or PpRopGAP1 rescued cell growth but resulted in increased and reduced cell width, respectively. Scale bar: 100 μm. **b** Quantification of cell width in WT, mutant, and rescued mosses. **c** Quantification of cell width in WT and overexpression lines. Data are presented as box-and-whisker plots, showing the interquartile range (box), the median (horizontal line), and minimum and maximum values (whiskers) in (**b**, **c**). The number of cells used for quantification is shown at the bottom. Each group is compared with WT cells using one-way ANOVA tests. ns, not significant. *p < 0.05. ****p < 0.0001. The exact *p*-values are available in Source Data. Source data are provided as a Source Data file.

ability of PpREN and PpRopGAP1 toward PpROP4DN (Supplementary Fig. 9c). Additionally, PpREN-PpROP4CA binding was slightly decreased and increased by deleting the PH domain (Δ1-162) and the CC motif-containing tail (Δ375-921), respectively. However, an arginine-to-leucine mutation (R204L) that blocks the GAP activity did not affect PpREN-PpROP4CA interaction. Therefore, the PpREN-PpROP4 binding appears to be regulated by intramolecular interactions and does not require GAP activity. Together, our data suggest that PpRopGAP1 has a stronger binding capacity to active PpROP4 than PpREN, and the binding ability of PpRopGAP1 and PpREN with active PpROP4 is differentially regulated by their noncatalytic domains.

## PpREN displays ten-fold higher GAP activity than PpRopGAP1

We next examined the GAP activity of PpRopGAP1 and PpREN in vitro. To our knowledge, the GTPase activity of PpROPs has not been tested to date. Therefore, we purified PpROP4 fused with polyhistidine (His) tags and examined its GTPase activity. Similar to Arabidopsis AtROP1[47], PpROP4 exhibited a dose-dependent intrinsic GTPase activity (Supplementary Fig. 10a). The addition of either MBP-PpRopGAP1 or MBP-AtRopGAP1 efficiently stimulated His-PpROP4- or His-AtROP1-catalyzed GTP hydrolysis (Fig. 8f; Supplementary Fig. 10b). Interestingly, although PpROP4 was more active than AtROP1 (Supplementary Fig. 10a), AtRopGAP1 exhibited relatively higher GAP activity than PpRopGAP1. Previously, the CRIB domain has been shown to enhance the GAP activity

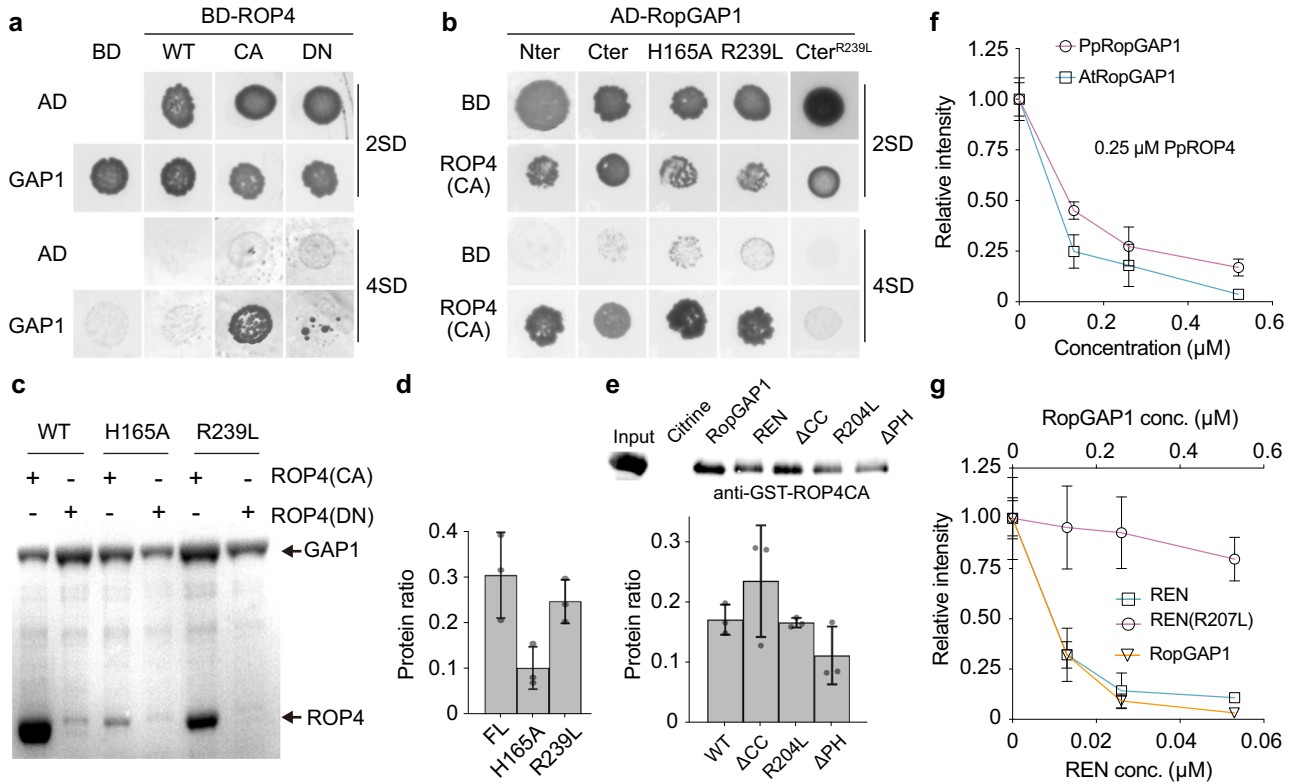

**Fig. 8 | PpRopGAP1 exhibits a higher binding capacity with PpROP4 but a much lower GAP activity than PpREN in vitro. a** PpRopGAP1 interacts with constitutively active (CA) PpROP4 but not the WT or dominant-negative (DN) forms in yeast-two-hybrid assays. 2 SD, nonselective Trp-/Leu- synthetic dropout medium. 4 SD, selective Trp-/Leu-/His-/Ade- medium. **b** The CRIB domain-containing N-terminal portion (Nter, 1-193) and the GAP domain-containing C-terminal portion (Cter, 194-469) of PpRopGAP1 can bind active PpROP4 separately. The binding of Cter with active PpROP4 is abolished by the activity-deficient R239L mutation. **c** Pull-down analyses of PpRopGAP1-PpROP4 interactions in vitro. Recombinant Glutathione S-transferase (GST)-fused PpROP4(CA) or PpROP4(DN) was purified and pulled down by maltose binding protein (MBP)-tagged WT or mutant PpRop-GAP1. Coomassie blue-stained SDS-PAGE gel after pull-down is shown. **d** Quantification of the relative amounts of GST-PpROP4(CA) pulled down by WT and mutant MBP-PpRopGAP1. **e** Pull-down analyses and quantification of PpREN-PpROP4CA interactions in vitro. GST-fused PpROP4(CA) was purified and pulled down by WT or mutant forms of MBP-PpREN. Citrine and RopGAP1 were used as negative control and positive control, respectively. ROP4CA was detected with an anti-GST antibody via western blotting. **f** The GTPase activity of PpROP4 stimulated by PpRopGAP1 and AtRopGAP1. Relative fluorescence intensity generated by excess GTP after GTPase reactions were measured. Each reaction contained 0.25 μM PpROP4. **g** GTPase activity of PpROP4 stimulated by PpREN, PpREN(R204L), and PpRopGAP1. Note that PpREN is highly active and can stimulate GTP hydrolysis at a concentration around ten-fold lower than PpRopGAP1. Each experiment in (**c–g**) was performed with three replicates. Data in (**d–g**) are presented as mean values ± SD with individual points shown in (**d, e**). Source data are provided as a Source Data file.

of RopGAPs in flowering plants[21,23]. We generated truncated forms of PpRopGAP1 and confirmed that the CRIB domain was also necessary for the full activity of PpRopGAP1 (Supplementary Fig. 10c). However, in contrast to the strong inhibition of GAP activity by the R239L mutation, the H165A mutation exhibited a mild effect on GTP hydrolysis (Supplementary Fig. 10d), demonstrating that the CRIB domain is not essential for but functions to promote GAP-stimulated GTP hydrolysis. Compared to PpRopGAP1, PpREN stimulated GTP hydrolysis by His-PpROP4 at a concentration about ten times lower (Fig. 8g). The GAP activity was severely reduced when the R204L mutation was introduced. Truncated proteins lacking the PH domain, CC motifs, or both did not significantly influence PpREN activity (Supplementary Fig. 11a). However, when the truncated proteins, as well as the R204L mutant protein, were expressed in *P. patens*, they failed to rescue cell growth defects (Supplementary Fig. 11b). Therefore, the PH domain, CC motifs, and GAP activity are required for PpREN function in vivo. Taken together, our results demonstrate that PpREN and PpRopGAP1 are bona fide GAPs but exhibit distinct biochemical activities.

### Overexpression of PpREN disrupts the membrane clustering of PpRopGAP1

We next asked how PpREN and PpRopGAP1 differently alter cell width. Since PpRopGAP1 delimits the apical membrane domain and the CRIB domain is necessary and sufficient for its membrane targeting, we reasoned that PpRopGAP1 was recruited to the membrane by active ROPs and formed clusters to limit ROP diffusion. Indeed, RopGAPs have been shown to form dimers in vitro[49] and are present in dispersive particles in vivo (Fig. 2a). Because PpRopGAP1 but not PpREN or its GAP domain was efficiently pulled down by GST-PpROP4CA (Fig. 9a), it appears unlikely that PpREN antagonizes PpRopGAP1 through competitive binding with PpROP4. However, when the Nter portion of PpRopGAP1 was fused with the GAP domain of PpREN, its membrane targeting ability was severely blocked (Fig. 9b, c). When PpREN was overexpressed, the endogenous PpRopGAP1 exhibited a similar reduction but to a lesser extent in membrane association (Fig. 9d, e). Noticeably, PpRopGAP1 decorated a broader region along the subapical membrane, although the mobility of PpRopGAP1-labeled particles was not affected (Fig. 9d–g). These data suggest that the expansion of the PpRopGAP1-labeled domain is not caused by changes in the diffusion ability or the assembly of PpRopGAP1 particles. Given the high activity of PpREN and the dependence of active ROPs in RopGAP1 recruitment, we concluded that the coalescence of PpRopGAP1 particles into large domains requires a substantial amount of active ROPs and relatively stable interactions between PpRopGAPs and PpROPs, which could be impeded by rapid inactivation of ROPs.

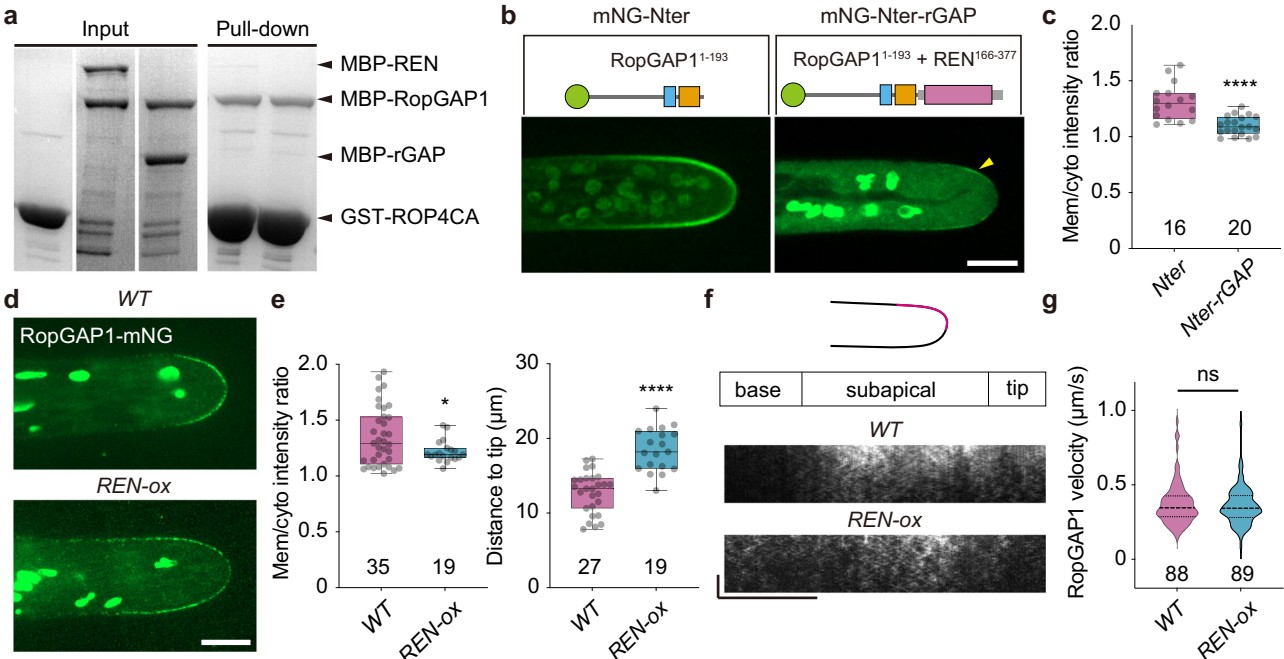

**Fig. 9 | PpREN disrupts the membrane clustering of PpRopGAP1. a** Competitive pull-down analyses of PpRopGAP1 and PpROP4CA in the presence of PpREN or the GAP domain of PpREN (rGAP). Equal amounts of MBP-PpRopGAP1 and MBP-PpREN (or MBP-rGAP) were pulled down by GST-PpROP4CA. **b** Localization of the free Nter portion of PpRopGAP1 and its fusion with the GAP domain of PpREN. The arrow indicates weak enrichment at the apical membrane. Scale bar: 10 μm.
**c** Quantification of intensity ratio between membrane-associated and cytoplasmic proteins. The number of cells used for quantification is shown below the bars.
**d** Localization of the endogenous PpRopGAP1 in WT and REN overexpression (REN-ox) lines. Scale bar: 10 μm. **e** Quantification of intensity ratio between membrane-associated and cytoplasmic PpRopGAP1 (left) and distances of PpRopGAP1 signal to the cell tip (right). The number of cells used for quantification is shown below the bars. **f** Kymographs showing PpRopGAP1 movement. Kymographs were generated

at one lateral surface along the tip to the basal region. Note that PpRopGAP1 is enriched at the subapical membrane in WT cells and its signal extends to the basal membrane in REN-ox cells. Scale bars: horizontal, 10 μm; vertical, 10 s.
**g** Quantification of velocities of PpRopGAP1-labeled particles. The numbers of mobile particles from WT (5 cells) and REN-ox (4 cells) cells are shown below the bars. Data in (**c**, **e**) are presented as box-and-whisker plots, showing the interquartile range (box), the median (horizontal line), minimum and maximum values (whiskers), and individual data points. Data in (**g**) are presented as violin plots, showing the quartiles (thin lines) and the median (central line). Statistical analyses were performed using two-tailed student's t-tests. ns, not significant. $*p < 0.05$. $****p < 0.0001$. The exact p-values are available in Source Data. Similar results in (**a**) were obtained in three independent experiments. Source data are provided as a Source Data file.

## Discussion

In this study, we report that PpRopGAPs and PpREN play overlapping roles in globally inactivating PpROPs and facilitating ROP accumulation in membrane domains during tip growth. Meanwhile, they have differential biochemical activities and act oppositely to regulate ROP organization and cell width. We propose that an optimal amount of active ROPs and proper clustering are necessary for tip growth and cell width regulation. However, cell width is more sensitive to the size of polar domains and is balanced by the antagonistic action of clustering-promoting RopGAPs and potent ROP inactivator RenGAPs (Fig. 10).

### The roles of global and local inhibitions of ROPs in polarity establishment

ROP-dependent polarity establishment has been well known to involve spatiotemporal control of ROP activity. In particular, local restriction of ROPs by RopGAPs is considered essential[23,50–52]. In this study, using complete knockout mutants, we unequivocally demonstrate that local inactivation of ROPs by RopGAPs is not essential for polarity formation (Fig. 4). By contrast, ROPs must be globally inactivated by PpRopGAPs and PpREN for efficient membrane enrichment (Fig. 6). Interestingly, even in the *ropgap, ren* mutants, polar localization of ROPs and tip growth are not completely abolished (Fig. 6a). As loss-of-function or overexpression of ROPs causes drastic growth and developmental defects[34–37,47,48,53,54], other regulators such as RopGDIs may also have overlapping roles with RopGAPs and RenGAPs. Indeed, RopGDIs have been reported to restrict polarity formation[55–57] and are partly redundant with RopGAPs in flowering plants[24]. If the local restriction of ROPs

is not essential for polarity initiation, how could ROPs segregate into polar membrane domains? One explanation is that globally inhibited ROPs could be locally activated by RopGEFs[19,58] and membrane lipids[18,48,59] and the activated ROPs are sufficient to assemble a polar membrane domain. This possibility is consistent with membrane cluster formation of ectopically expressed RopGEFs and ROPs in the absence of RopGAPs[51]. More importantly, we observed a much earlier recruitment of PpROP4 and PpRopGEF4 in transient domains than PpRopGAP1 (Fig. 2), implying that RopGAPs are not required for polar domain initiation. Then how do ROP-related GAPs contribute to polarity establishment? One major function of GAPs is to maintain an optimal level of active ROPs, which is important for restricting polarity initiation and preventing overactivation-induced depolarization. This is accomplished by the synergistic action of RopGAPs and RenGAPs to globally inactivate ROPs (Figs. 4, 5). Another function of GAPs is to set up an optimally restricted polar membrane domain for cell width regulation. This is triggered by the recruitment of RopGAPs to cluster ROPs into a confined region, a process that depends on relatively stable interactions between RopGAPs and active ROPs. On the other hand, RenGAPs rapidly inactivate ROPs and counteract with RopGAPs in generating confined membrane domains, thus providing an elaborate mechanism to finetune cell size.

### Functional conservation and diversification of RopGAPs and RenGAPs

RopGAPs and RenGAPs are exclusively found in land plants[20] and green algae *K. nitens*, suggesting that they emerged around the time of the

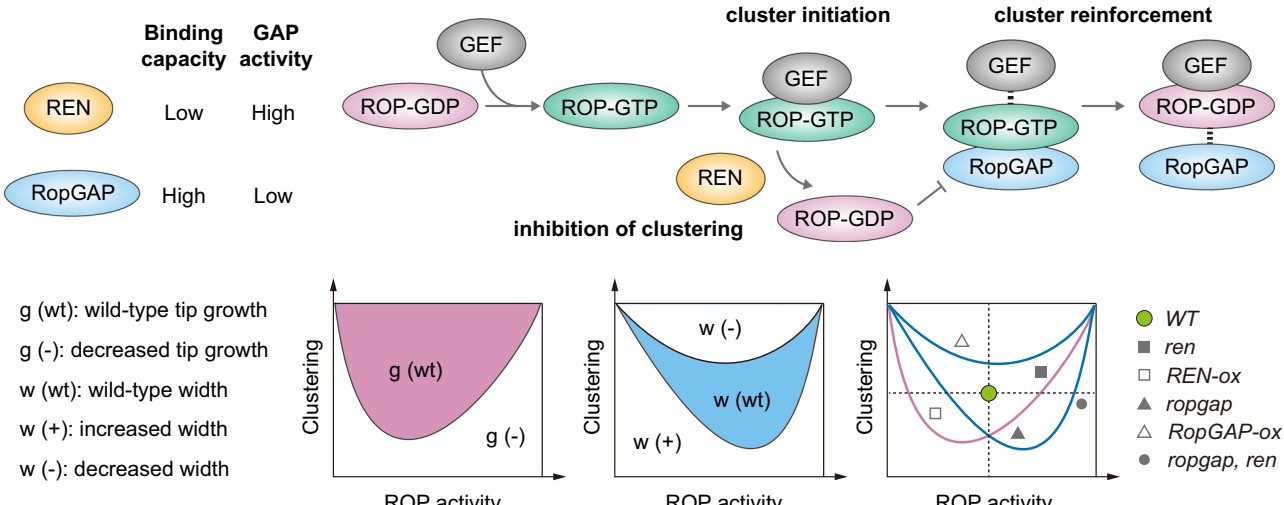

**Fig. 10 | A working model on the functions of REN and RopGAP in coordinating tip growth and cell width regulation.** REN has a higher GAP activity but a weaker binding capacity to active ROPs than RopGAPs. During polarity establishment, RopGEFs convert ROP-GDP (inactive) to ROP-GTP (active) and drive polarity initiation by forming initial clusters with ROPs. RopGAPs are subsequently recruited by active ROPs and promote the formation of large clusters owing to their relatively stable interactions with ROPs, leading to a stabilized and restricted polar membrane domain. REN stimulates the rapid conversion of ROP-GTP to ROP-GDP, thus decreasing the amount of active ROPs and weakening the clustering ability of RopGAPs. Tip growth and cell width are both regulated by the level of active ROPs and their clustering capacity but display distinct sensitivities to these two parameters. Alterations in ROP activity and clustering by perturbing the functions of RopGAPs and REN could lead to different effects on tip growth and cell width.

colonization of land by plants or earlier. However, the presence of distinct regulatory domains as well as biochemical characteristics indicates that they may be regulated in different ways. Indeed, the CRIB domain in RopGAPs enhances the GAP activity[21,23] and targets RopGAPs to the membrane by interacting with active ROPs (Fig. 3). By contrast, the activity of RenGAPs appears to be not influenced by the PH domain or the CC tail domain (Supplementary Fig. 11a)[22]. An early study reveals a possible role of the CC tail of AtREN1 in association with exocytic vesicles[22]. Recently, AtPHGAPs have been shown to be stabilized by phosphorylation in the CC tail domain, which consequently inactivates AtROP2 at the indenting regions in pavement cells[60,61]. RenGAPs could also potentially associate with the membrane and vesicles via the lipid-binding PH domain[26]. However, whether this mechanism is widely conserved requires further investigation because the membrane association ability of RenGAPs in moss protonema cells (Fig. 4g), pollen tubes[22], and root hairs[26] seems to be different.

Even within the same subfamilies, sequence and functional analyses suggest diversifications. For instance, the PCRIB motif is absent in two subclades including AtRopGAP5 (Supplementary Figs. 2, 3), implying that some members of RopGAPs may employ different mechanisms for regulation. In barley *Hordeum vulgare*, HvMAGAP1 localizes to microtubules through its C-terminal tail and cooperates with HvRACB, a ROP GTPase family member, to regulate microtubule organization during fungal infection[62,63]. Recent studies have also revealed diverse functions of RenGAPs in Arabidopsis. AtREN2/AtPH-GAP1 and AtREN3/AtPHGAP2 are located at the CDZ and regulate division plane orientation in root cells[64]. In pavement cells, they negatively regulate AtROP2 on the anticlinal surface of indenting regions but are degraded by brassinosteroid signaling in the lobe to promote interdigitation[60,61]. In contrast, AtREN1 associates with secretory vesicles and controls pollen tube growth by globally inactivating ROPs[22]. *P. patens* has only one RenGAP PpREN. Although it plays a similar role in tip growth to AtREN1, it also labels the CDZ and this localization requires the CC motif-containing tail but not the PH domain or GAP activity (Supplementary Fig. 11c). However, we did not reveal obvious defects in division plane orientation in the *ren* or *rop-gap, ren* mutants. In addition, none of the PpRopGAPs was found at the CDZ, thus excluding possible genetic redundancy between PpREN and

PpRopGAPs. Whether PpRopGAPs and PpREN play a direct role in cell division remains an open question.

## Methods

### Moss strains and culture conditions
All strains used in this study were derived from the Gransden ecotype of *Physcomitrium patens*. Mosses were routinely grown on standard solid BCDAT medium under continuous white light at 23 - 25 °C. For colony growth analyses, approximately equal amounts of protonema tissues were inoculated on BCDAT plates and cultured for two weeks. Colonies were photographed every two days with a digital camera. Mosses used for high-resolution imaging were cultured for 5–7 days on a thin layer of BCD medium in 35 mm imaging dishes.

### Sequence alignment and phylogenetic analyses
*P. patens* genes encoding RopGAPs, RenGAPs, and RopGEFs were characterized by BLASTing the Arabidopsis homologs against the *P. patens* genome v3.3 in the Phytozome database (https://phytozome-next.jgi.doe.gov). A total of six RopGAPs, one RenGAP, and six Rop-GEFs were characterized, which was consistent with previous reports[20,38]. To isolate RopGAPs in other species, protein isoforms carrying a CRIB domain (IPR000095) and a RhoGAP domain (IPR000198) in the InterPro database (https://www.ebi.ac.uk/interpro/search/ida) were retrieved through domain architecture search. A total of 1 276 sequences were isolated. 1 244 of them were successfully mapped in the UniProt database (http://www.uniprot.org) and used for alignment and phylogenic analyses. Alignment was performed with the Clustal Omega service in Jalview (https://www.jalview.org). Phylogenetic trees were constructed by MEGA11 (https://www.megasoftware.net) using the neighbor-joining method with a 1 000 bootstrap value. Trees were rendered with the iTOL tool (https://itol.embl.de). Domain characterization was performed with the Conserved-Domain Search service from National Center for Biotechnology Information (https://www.ncbi.nlm.nih.gov/Structure/cdd/wrpsb.cgi). Among the 1 244 sequences, 1 063 were characterized to contain a conserved pre-CRIB motif preceding the core CRIB motif. Homologs of RenGAPs were isolated similarly by searching for sequences carrying a pleckstrin homology domain (IPR001849), a RhoGAP domain (IPR000198), and a

ternary complex factor MIP1-leucine-zipper domain (IPR025757). A total of 985 sequences were isolated. Homologs of RopGAPs and RenGAPs were found in land plants and the green algae *Klebsormidium nitens* but not in other available genomes of plants including red algae and other green algae in the InterPro or Uniprot databases.

## Plasmid construction

For homologous recombination-mediated knockout of PpREN and knock-in of PpROP4, PpREN, PpRopGAPs, and PpRopGEF4, genomic sequences at the length of 700 - 2000 bp were amplified from genomic DNAs with PrimeStar Max DNA polymerase (Takara, cat. R045A). The resulting fragments were cloned into vector backbones derived from pPY22 (mNeonGreen-Hygromycin B) or pPY23 (mNeonGreen-Nourseothricin) using the In-Fusion Snap Assembly Kit (Takara, cat. 638948). For PpREN knockout, the entire coding region was replaced by a Nourseothricin-resistance gene expression cassette. For N-terminal knock-in, the codon-optimized mNeonGreen sequence[36] was inserted to the 3' of the start codon of PpRopGAPs. For C-terminal knock-in, the mNeonGreen (or Citrine)-Hygromycin B or mCherry-Hygromycin B cassette was inserted to the 3' end of the coding region of PpRopGAP1 or PpRopGEF4, respectively. For internal tagging of PpROP4, the mNeonGreen sequence was inserted between the Gly134 and Ala135[35]. Overexpression plasmids were constructed as follows: First, the coding sequences were cloned into a pENTR plasmid in-frame with an N-terminal Cerulean (for rescue experiments) or mNG (for localization analysis) by In-Fusion Snap Assembly. The resulting fusion protein-coding fragment was then inserted into the vector pPY138 via LR reactions (Gateway LR Clonase Enzyme Kit, Thermo, cat. 11791019) for constitutive overexpression under the control of EF1α promoter. Intron-free coding sequences used in this study were amplified from total cDNAs reverse transcribed using the Hifair III 1st Strand cDNA Synthesis Kit (Yeason, cat. 11139ES10). The sequences were inserted into pGBKT7/pGADT7 vectors for yeast-two-hybrid analyses or into pGEX-GST/pET28a-MBP vectors for protein expression by In-Fusion Snap Assembly. Mutations were introduced by PCR amplification of plasmids with mutation-carrying primers and subsequent transformation into DH5α bacteria. In the Phytozome database, there are three isoforms of PpREN. Our cDNA amplification isolated the shortest one that comprises 921 amino acids and lacks several unconserved residues in the PH domain.

For CRISPR/Cas9 knockout of PpRopGAPs, single-guide RNA (sgRNA) target sequences (Supplementary Fig. 6) were selected at conserved regions in the CRIB domain or GAP domain. They were synthesized as oligonucleotides and annealed to form double-stranded DNAs carrying overhangs. Subsequently, the DNAs were inserted into the plasmid pPY156, which carried a Hygromycin B-resistance gene expression cassette, between the pU6 promoter and sgRNA scaffold via *Bsa* I digestion and T4 DNA ligase-mediated ligation (Takara, cat. 6023). For oligonucleotide-dependent mutation knock-in, the templates (shown in Supplementary Fig. 6) were synthesized as oligonucleotides as described before[45]. Detailed information about the plasmids used in this study is available in Supplementary Table 1.

## Moss transgenesis

Our moss transformation followed the previously published protocol[65]. In brief, the moss protonemata were collected and digested with driselase to obtain protoplasts. The protoplasts were transformed with 30 μg linearized plasmids for homologous recombination. If the homologous templates did not carry an antibiotics-selection marker, an appropriate antibiotics-resistance plasmid was cotransformed. In the CRISPR/Cas9 knockout experiments, 10 μg Cas9-expressing plasmids and 10 μg sgRNA plasmids were cotransformed. For oligonucleotide-dependent mutation knock-in, 10 μL of 50 μM single-stranded oligonucleotides were cotransformed together with the Cas9 and sgRNA plasmids. Transformed protoplasts were plated on

cellophane-laid PRM plates and cultured for 10-14 days during which the cellophane disks were transferred to selection plates once for transient selection of CRISPR/Cas9 knockout lines or N-terminal tagging lines and twice for selection of C-terminal tagging lines, overexpression lines, or PpREN knockout lines, respectively. Positive colonies were propagated and verified by PCR-amplifying the inserted fragments with KOD-ONE DNA polymerase (Toyobo, cat. KMM-101).

## Image acquisition, processing, and analysis

Protonema tissues cultured in BCD medium were imaged using an Axio Observer Z1 spinning-disk confocal microscope (Zeiss) equipped with a Yokogawa confocal spinning disk unit, a 63 × 1.40 NA or 100 × 1.40 NA oil-immersion objective lens, and a Hamamatsu camera. The excitation/emission wavelengths for green and red fluorescence imaging were 488/517 nm and 561/603 nm, respectively. Time-lapse imaging of cell growth or division was performed at an interval of two or three minutes. An interval of 500 ms was used for imaging of PpRopGAP1 dynamics on the membrane. FM4-64 staining was carried out by adding 10 μM of FM4-64 diluted in water to imaging samples and keeping the samples in the dark for 30 min. The dye solution was subsequently removed before confocal imaging. Images were acquired using Zeiss Zen software (Version 2.3, Blue Edition) and processed and analyzed with Fiji software (Version 2.14.0). As tip cells can vary in length, the quantification of cell length was performed using the subapical cells. Similarly, cell width was measured at the subapical region to ensure consistency. To visualize fluorescence distribution at the apical membrane, a segmented line with spline fit was manually drawn along the apical membrane to cover the entire dome region. The corresponding intensity profile was subsequently generated. The enrichment of PpROP4 at the apical membrane was quantified by calculating the ratio of peak intensity to base intensity in the profile. The membrane association of PpRopGAP1 or its Nter portion was measured by normalizing fluorescence intensity at a manually selected flanking region of the cell apex to a nearby area in the cytoplasm of the same size.

## Yeast-two-hybrid analyses

Approximately 0.1 μg of plasmids that express the AD- or BD-fusion proteins were cotransformed with 100 μg carrier DNAs into competent cells of the AH109 strain. Transformed cells were cultured on a solid YPDA medium without Trp and Leu supplements for selection. Positive colonies were propagated and subcultured on a solid YPDA medium without Trp, Leu, His, and Ade supplements to test protein interactions.

## Protein expression, purification, and pull-down assays

Proteins were expressed in 40 ml of the BL21(DE3) bacteria and harvested in lysis buffers containing 50 mM Tris-HCl (pH 7.5), 300 mM NaCl, 1% NP-40, 1 mM PMSF, and 1 mM DTT. The TALON Metal Affinity Resin (Takara, cat. 635501), Pierce Glutathione Agarose (Thermo, cat. 16100), and Amylose Resin (New England Biolabs, cat. E9021V) were used for purifying proteins fused with polyhistidine (His), Glutathione S-transferase (GST), and maltose binding protein (MBP) tags, respectively, following the manufacturer's instructions. His-, GST-, and MBP-tagged Proteins were eluted with 300 mM imidazole, 10 mM glutathione, and 10 mM maltose, respectively, in 20–50 mM Tris-HCl pH 7.5 solutions. In the pull-down experiments, equal amounts of purified constitutively active (G15V) or dominant-negative (D20N) forms of PpROP4 were incubated with Amylose resins loaded with wild-type or mutated MBP-PpRopGAP1/MBP-PpREN for four hours. MBP-Citrine was used as a negative control. Resins were washed at least three times and then boiled. Denatured proteins were separated on polyacrylamide gels. For the PpRopGAP1 pull-down, GST-PpROP4 was directly stained with Coomassie blue for analysis. For PpREN pull-down, GST-PpROP4 was detected with an anti-GST primary antibody

(26H1) of mouse origin (Cell Signaling Technology, cat. 2624-100 µl, 5000 × dilution) and an anti-mouse-HRP (goat) secondary antibody (Sigma, cat. A4416-1 mL, 2000 × dilution) by western blotting. Hemagglutinin (HA)-tagged mutant PpRopGAP1 in yeasts was detected with an anti-HA primary antibody (C29F4) of rabbit origin (Cell Signaling Technology, cat. 3724 T, 5000× dilution) and an anti-rabbit-HRP (goat) secondary antibody (Cell Signaling Technology, cat. 7074P2, 2000 × dilution).

## In vitro GTPase assays

Before GTPase reactions, proteins were concentrated by ultra-centrifugation in Ultrafiltration Spin Columns (Beyotime, cat. FUF051). The concentration of each protein was determined by running a parallel BSA gradient with 1–5 µL of purified protein on the polyacrylamide gel. GTPase assays were performed using the GTPase-Glo Assay Kit (Promega, cat. V7681) basically following the manufacturer's instructions. In brief, each reaction was carried out in a 16 µL solution, which contained 5 µM GTP and appropriate amounts of ROP GTPases and/or GAPs. To measure the intrinsic GTPase activity of His-PpROP4 and His-AtROP1, 0.5 µg (1.3 µM), 1.0 µg (2.5 µM), and 5.0 µg (12.5 µM) of each protein were added to the reaction mixture, and the GTPase reaction was performed at room temperature for 1 h. After that, 16 µL of reconstituted GTPase-Glo reagent was added and mixed. The mixture was kept at room temperature for 30 min. Subsequently, the detection reagent was added and incubated for 5–10 min. Then fluorescence signals were detected with a microplate reader (BioTek Synergy H1). To measure GAP-stimulated GTPase activity, 0.1 µg (0.25 µM) of His-PpROP4 or His-AtROP1 was used for each reaction. 0.2 µg (0.13 µM), 0.4 µg (0.26 µM), and 0.8 µg (0.52 µM) of MBP-PpRopGAP1 or MBP-AtRopGAP1 were tested for GAP activity. 30 ng (0.013 µM), 60 ng (0.026 µM), and 120 ng (0.052 µM) of MBP-PpREN were tested for GAP activity. The GAP-stimulated GTPase reaction was performed at room temperature for 30 min. Each reaction was carried out with three replicates. The intensity was normalized to controls tested with purified MBP alone. To compare the GAP activity of wild-type and mutant forms of MBP-PpRopGAP1 or MBP-PpREN, an equal molar ratio of each protein was used as shown in the figure legends.

## Statistical analyses

All sample sizes for analysis were indicated in the figure or figure legends. Data were presented as mean ± SD or box-and-whisker plots. Statistical analyses were performed using two-tailed student's t-tests for comparing two groups or adjusted one-way ANOVA for comparing multiple groups. A significant difference between groups was determined when the p-value was less than 0.05.

## Accession numbers

PpRopGAP1 (Pp3c4_16800); PpRopGAP2 (Pp3c3_5940); PpRopGAP3 (Pp3c13_4010); PpRopGAP4 (Pp3c4_24980); PpRopGAP5 (Pp3c26_4490); PpRopGAP6 (Pp3c26_5960); PpREN (Pp3c9_17460); PpRopGEF4 (Pp3c2_28420); PpROP4 (Pp3c10_4950); AtROP1 (AT3G51300); AtRopGAP1 (AT5G22400).

## Reporting summary

Further information on research design is available in the Nature Portfolio Reporting Summary linked to this article.

## Data availability

All relevant data are available within the manuscript and its supplementary materials. Source data are provided with this paper.

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

## Acknowledgements

We thank Dr. Gohta Goshima for providing some of the plasmids, reagents, and moss strains. The original plasmids used for yeast-two-hybrid analyses and protein expression and some of our primary antibodies were gifts from Dr. Xingguang Deng, Dr. Dawei Zhang, and Dr. Kefeng Lu. We are grateful to Dr. Honghui Lin, Dr. Huapeng Zhou, Dr.

Zhonghan Li, Jingjing Sha, and Xiaohui Liu for their kind technical support. This project was supported by the National Natural Science Foundation of China (32270776), Sichuan University (1082204112609, 2022SCUH0010, 0082604151432), and the People's Government of Sichuan Province to P.Y.

## Author contributions
P.Y. conceptualized this project. J.R. and P.Y. conducted moss experiments. L.L. performed biochemistry experiments. L.L. and H.O. conducted yeast-two-hybrid assays. J.R., L.L., and P.Y. analyzed the data. P.Y. wrote the manuscript.

## Competing interests
The authors declare no competing interests.
