## [Peer Review File · Nature Communications]

Two subtypes of GTPase-activating proteins coordinate tip growth and cell size regulation in *Physcomitrium patens*REVIEWER COMMENTS

Reviewer #1 (Remarks to the Author):

The submitted manuscript by Ruan et al. describes the functions of two types of RopGAP members as regulators of polar growth, using *Physcomitrium patens* as model system. The authors employ a broad portfolio of techniques, including microscopic localization, yeast-2-hybrid and pull-down interaction assays, mutant phenotyping (incl. the generation of sextuple RopGAP and septuple RopGAP/RenGAP mutant lines), as well as in vitro GAP assays. The study provides convincing evidence for partially redundant GAP functionality with both GAP types exhibiting distinct subcellular localization, different activity levels and ROP-binding affinities. The authors develop a model to describe the coordination of cell growth and width through an interplay of ROP activity and clustering, which are modulated by GAP activities.

Overall, the study is carefully executed, the paper is very well written, figures are of top quality, and statements are generally cautiously phrased. The only point I am not fully convinced by is the proposed role of PpRopGAP1 in cell size regulation, as I am missing direct evidence (see comment below). I am also missing a control experiment that tests whether the fluorescently labeled GAPs are indeed functional, hence representing true subcellular localization. However, I believe my comments are all addressable by the authors.

Specific comments:

Fig 1/ Fig 4: The localization data of the various fusion proteins is generally of high quality. However, it is critical to test the functionality of the FP fusions. It is not clear if the rescue experiments shown in Fig 4 and later have been performed with the constructs carrying FPs or with unlabelled versions. If the latter is the case, please include experiments that test if FP fusion proteins are also rescuing the phenotypes in the septuple mutant.

l. 227: What is meant by „is required but not essential“? Consider rephrasing.

Fig 5/l. 251 ff.: The statement that PpREN and PpRopGAPs limit the size of the bulge appears too strong considering the data presented. The observation that the overexpression of PpRopGAP1 inhibits bulge formation could also be explained by an overall inhibition of polar growth. Fig 5A indicates that branching is only delayed but not abolished upon overexpression of PpRopGAP1. It would be interesting to know if and how the width of those bulges is altered.

l. 306-308: The negative Y2H data for PpROP4CA and PpREN/GAP domain appears to be missing from the manuscript. Please include the data in the supplement.

Fig 3c: the legend is not mentioning the used assay (yeast two hybrid). Please add this information.

Fig 3d: in order to test whether the observed low protein amounts for PpRopGAP1(R134/135L) are indeed caused by low protein stability, mRNA levels should be measured (e.g. via qRT-PCR) in parallel.

Fig 3e: suggestion: in order to allow for a more quantitative comparison of expression levels, would it be possible to use the chloroplast autofluorescence as a means of normalization?

Fig 4e: were the rescue experiments only performed in the septuple mutant background? This is not entirely clear.

Fig 4h: it is not clear to me how the classification of localization and rescuing ability was

performed. Please explain.

Fig 7: If possible, please replace the top-right and bottom-left panels with bright field images having less background particles. It is challenging to assess the cell morphology.

Reviewer #2 (Remarks to the Author):

In the Manuscript „Two subtypes of GTPase-activating proteins coordinate tip growth and cell size regulation in *Physcomitrium patens*” Ruan et al. show convincingly that two distinct types of GTPase-activating proteins (GAPs) regulate ROP signalling in protonema cells of *Physcomitrium patens*. The exciting and novel aspects of this manuscript are that both types of GAPs act, on the one hand, very redundant, as only a knockout of all GAPs from both types leads to strong phenotypes. On the other hand, cellular localization and biochemical properties of both GAP types are very distinct and lead to specific modulation of ROP function and, thus, cellular growth. Additionally, the authors describe a novel domain in RopGAPs that is required for ROP binding and clustering of the complex in the plasma membrane, which leads to the clustering of ROP signalling complexes at the plasma membrane.

This manuscript is well-written, convincing and represents original research that is of major significance to the field and our understanding of ROP signalling and the complex formation of signalling components at the plasma membrane. Therefore, this manuscript would be of major interest to this field. Most experiments and shown results are of high quality, support the authors' conclusions and are sufficiently detailed.

In conclusion, the authors provide a well-written manuscript with convincing results that allow the presented conclusions. Therefore, I suggest the publication of this manuscript in Nature Communications after a few issues are addressed.

Major:

Line 80-82: the authors conclude that RopGAPs, GEFs and ROPs are organized in clusters, which is convincing from the shown images and measurements. However, the authors do not convincingly show that the observed clusters are the very same for all three proteins. As the authors possess lines with RopGAP one coexpressed with either ROP4 or GEF4 this should be included in Figure 2.

Line 123-125: The authors use alpha fold to show that RopGAPs have an α -helical domain (PCRIB) preceding the CRIB domain. If this domain is crucial for CRIB folding and function, how is this achieved in other CRIB-domain proteins, like RICs or AtGAP5? The authors should include alpha-fold structures of such proteins, if these proteins may possess a similar fold with a different protein sequence.

A major point which needs improvement is the protein-protein interaction analysis. The Y2H results are surprising as one would expect interaction of RopGAP1 not just with the CA-ROP4 but also with the WT version, as active ROPs will also be present in yeast. Furthermore, there is slight growth of yeast in the RopGAP1 – DN-ROP4 combination, suggesting weak interaction between the two. This is also visible in the pull-down experiments, where the authors do not comment on the presence of ROP4 in the WT and H165A versions of RopGAP1. This interaction with inactive ROP might be weak, but it seems stronger than this RenGAP to ROP4. Such an interaction of RopGAP to active and inactive ROPs would explain why RopGAPs behave differently and form clusters with ROPs in active ROP signalling domains. The interaction strength might be different, but it would

allow RopGAPs to stay in a complex with ROPs after inactivation and thus would be crucial for persistent complex formation at the plasma membrane. Therefore, it would be crucial to show to what extent RopGAPs interact with DN-ROP and how the interaction strength differs from CA-ROP or RenGAP-ROP interaction. To make such interaction studies convincing and quantitative MST experiments (or similar quantifiable approaches) should be performed. If this is technically not possible, at least the interaction of RopGAPs o all activity forms of ROPs should be compared also with Western blot after the pull-down and compared to the presented RenGAP-ROP interaction.

Minor:

Abstract: The first two sentences should be revisited, as cell polarity is not a mechanism. Cell polarity results from mechanisms leading to concentration differences of cellular components, thus, breaking symmetry.

Line 4: "in higher plants". This should be rephrased to seed plants (spermatophytes). The height of the plants is irrelevant.

Line 17-19: When introducing and discussing ROP clustering at the plasma membrane, the authors should include the work of Alexandre Martinière and Yvon Jaillais. Especially Platre et al. Science 2019 (DOI: 10.1126/science.aav9959) and Smokvarska et al. Current Biology 2020 (DOI:10.1016/j.cub.2020.09.013).

Line 39: "basal land plants". The authors should rephrase this. *P. patens* might have a less complex body architecture than other land plants, but it is still a complex organism that evolved for the same time as any other existing organism and is thus not "basal".

Line 105: Include the construct details (amino acid 1-193, including PCRIB and CRIB domain) also in the main text for easier understanding.

Line 137: "Expression in vitro". It is unclear which systems the authors are describing. Was an in vitro translation system used and data is not shown, or do the authors refer to the expression in *E. coli*?

Line 162: "expression is completely blocked". The authors do not test expression (mRNA production and presence) and should thus rephrase this.

Line 191: "local inactivation of ROPs by RopGAPs is required". It is unclear why one can conclude that local inactivation is required. It might be that the overall reduction of RopGAP activity causes the mild phenotypes of the *ropgap* mutant. The authors should clarify this.

Line 199: "In *ropgap* or *ren* mutants alone". Unclear reference to genotype. Rephrase if single mutants or which genotype is meant.

Line 230: "we frequently observed early accumulation" should be rephrased, as this was observed in all cells.

Line 61: "found exclusively in land plants and *K. nitens*". The authors should consider that our knowledge about the genomes of other streptophyte algae is very limited. If RopGAPs and RenGAPs are found in *k. nitens*, it is very likely that they will also be present in other Streptophyte algae and thus have an earlier evolutionary origin than suggested by the authors.

Point-to-point responses to the reviewers' comments

We thank the reviewers for their constructive comments. In the revised manuscript, we have made several improvements as suggested:

First, we performed three sets of experiments and added new data. The data includes:

- (1) mRNA expression levels of wild-type and mutant RopGAP1 detected by qRT-PCR analyses (Figure S5).
- (2) A comparison of binding abilities of RopGAP1 and REN with different activity forms of ROP4 detected by pull-down and western blotting analyses (Figure S9).
- (3) RopGAP1-RopGEF4 and ROP4-RopGEF4 colocalization in membrane particles examined via time-lapse imaging of double-labeled lines (Figure 2G).

Second, we performed additional analyses, including:

- (1) identifying a PCRIb-like helical motif in RICs, i.e. CRIB domain-only proteins (Figure S4).
- (2) quantifying the size of bulges in RopGAP1-overexpression cells (Figure 5C).

Third, we rephrased some interpretations and improved the figures as suggested by the reviewers.

Our point-to-point responses are shown below, highlighted in blue. The rephrased parts in the manuscript are also shown below, italicized and colored brown for the reviewers' reference.

Reviewer #1 (Remarks to the Author):

The submitted manuscript by Ruan et al. describes the functions of two types of RopGAP members as regulators of polar growth, using *Physcomitrium patens* as model system. The authors employ a broad portfolio of techniques, including microscopic localization, yeast-2-hybrid and pull-down interaction assays, mutant phenotyping (incl. the generation of sextuple RopGAP and septuple RopGAP/RenGAP mutant lines), as well as in vitro GAP assays. The study provides convincing evidence for partially redundant GAP functionality with both GAP types exhibiting distinct subcellular localization, different activity levels and ROP-binding affinities. The authors develop a model to describe the coordination of cell growth and width through an interplay of ROP activity and clustering, which are modulated by GAP activities.

Overall, the study is carefully executed, the paper is very well written, figures are of top quality,

and statements are generally cautiously phrased. The only point I am not fully convinced by is the proposed role of PpRopGAP1 in cell size regulation, as I am missing direct evidence (see comment below). I am also missing a control experiment that tests whether the fluorescently labeled GAPs are indeed functional, hence representing true subcellular localization. However, I believe my comments are all addressable by the authors.

We appreciate the reviewer's positive and insightful comments and have addressed the reviewer's concerns below.

Specific comments:

Fig 1/Fig 4: The localization data of the various fusion proteins is generally of high quality. However, it is critical to test the functionality of the FP fusions. It is not clear if the rescue experiments shown in Fig 4 and later have been performed with the constructs carrying FPs or with unlabelled versions. If the latter is the case, please include experiments that test if FP fusion proteins are also rescuing the phenotypes in the septuple mutant.

We apologize for the unclarity. Our rescue experiments were all performed with Cerulean or mNeonGreen fusion proteins. This information is now included in the main text as well as in the legend of Figure 4A and the Materials and Methods section.

Main text: *"... rescued by overexpressing PpREN or PpRopGAP1 fused with an N-terminal Cerulean."*

Legend of Figure 4A: *"Cerulean-fusion PpREN or PpRopGAP1 (simplified as + REN and + GAP1) were expressed for rescue in the ropgap, ren septuple mutant background."*

Materials and Methods: *"Overexpression plasmids were constructed as follows: First, the coding sequences were cloned into a pENTR plasmid in-frame with an N-terminal Cerulean (for rescue experiments) or mNG (for localization analysis) by In-Fusion Snap Assembly."*

I. 227: What is meant by "is required but not essential"? Consider rephrasing.

We deleted the words "is required" because the CRIB domain not only targets RopGAPs to the membrane, thus inactivating ROPs locally but may also globally enhance GAP activity. It is safer to not conclude local inactivation is specifically required for polarity formation.

"local inactivation of ROPs by RopGAPs is not essential for polarity formation and tip growth."

Fig 5/I. 251 ff.: The statement that PpREN and PpRopGAPs limit the size of the bulge appears too strong considering the data presented. The observation that the overexpression of PpRopGAP1 inhibits bulge formation could also be explained by an overall inhibition of polar growth. Fig 5A indicates that branching is only delayed but not abolished upon overexpression of PpRopGAP1. It would be interesting to know if and how the width of those bulges is altered.

We agree with the reviewer that the overall inhibition of polar growth may also affect bulge formation. As suggested, we quantified the width and height of bulges in PpRopGAP1 overexpression lines. However, we did not find a significant difference from those of WT cells (Figure 5C). In addition, PpRopGAP1 overexpression could fully rescue cell length phenotypes (Figure 4E), suggesting that tip growth is not affected. In the *ropgap, ren* septuple mutant, the apical growth of bulges (i.e. bulge height in Figure 5C) is not affected either, even though the growth of tip cells is reduced (Figure 4E and 4F). It appears that branch initiation and polar growth exhibit different sensitivities to changes in ROP activity, with the former more sensitive to ROP inactivation (e.g. induced by PpRopGAP1 overexpression) and the latter more sensitive to ROP hyperactivation (e.g. induced by *ropgap, ren* loss of function). Moreover, we have previously shown that globally reduced polar growth does not necessarily induce branching defects (Yi and Goshima, *Curr. Biol.*, doi: 10.1016/j.cub.2020.05.022). Nevertheless, the change of cell width in the *ropgap, ren* septuple mutant cannot completely rule out the involvement of growth-regulating functions of PpREN and PpRopGAPs in branch patterning. Therefore, we toned down our statements and concluded that PpREN and PpRopGAPs function to inhibit branch initiation and regulate bulge morphology.

“We concluded that PpREN and PpRopGAPs are not essential for phragmoplast guidance but could indirectly influence cell division by altering bulge morphology. In support of this notion, PpRopGAP1 overexpression strongly delayed bulge emergence (Figure 5A) and could lead to division plane mispositioning when the cells failed to initiate bulges (Figure 5F). Such phenotypes are also observed in rop mutants or cells treated with actin inhibitors, supporting that PpRopGAPs, PpROPs, and actin may function in a common pathway. Interestingly, the width and height of bulges in PpRopGAP1 overexpression lines were not significantly different from those of WT cells (Figure 5C). Therefore branching defects are likely caused by reduced polarity initiation rather than by global inhibition of polar cell growth. This notion is in line with the facts that PpRopGAP1 overexpression does not markedly affect cell length (Figure 4F) and the apical growth of bulges (i.e. height in Figure 5C) is normal in the ropgap, ren septuple mutant. Taken together, these results indicate that PpREN and PpRopGAPs function to restrict branch initiation and suggest that polarity initiation and polar cell growth have different sensitivities to ROP activity, with the former more sensitive to ROP inactivation and the latter more sensitive to ROP hyperactivation.”

I. 306-308: The negative Y2H data for PpROP4CA and PpREN/GAP domain appears to be

missing from the manuscript. Please include the data in the supplement.

We have included the Y2H data in the new Figure S9B.

Fig 3c: the legend is not mentioning the used assay (yeast two-hybrid). Please add this information.

We have added this information to the legend of Figure 3C.

“The PCRI_B motif facilitates the interaction between the full-length (FL) PpRopGAP1 and constitutively active PpROP4 (ROP4CA) in yeast-two-hybrid assays.”

Fig 3d: in order to test whether the observed low protein amounts for PpRopGAP1(R134/135L) are indeed caused by low protein stability, mRNA levels should be measured (e.g. via qRT-PCR) in parallel.

We have performed qRT-PCR analyses as suggested and found no significant difference among the groups. The data is provided in the new Figure S5 and the results are mentioned in the main text as follows.

“Indeed, the R134/135L double-mutant protein was poorly expressed in contrast to those carrying one mutation (Figure 3D), although the expression at the transcription level was not affected (Figure S5).”

Fig 3e: suggestion: in order to allow for a more quantitative comparison of expression levels, would it be possible to use the chloroplast autofluorescence as a means of normalization?

We thank the reviewer for the kind suggestion. In our experience, the autofluorescence of each chloroplast can be very variable in tip cells. Therefore, we do not think that it is a good reference for normalization. Anyway, membrane localization can be visually and empirically distinguished from complete cytosolic localization, even in the presence of relatively low intensity. To describe the localization patterns more qualitatively, we added the number of cells that exhibit membrane localization on each representative image and mentioned this in the legend of Figure 3E.

“The number of cells with a membrane localization pattern is shown.”

Fig 4e: were the rescue experiments only performed in the septuple mutant background? This is not entirely clear.

Yes. Our rescue experiments were all performed in the septuple mutants. We have added this information to the legend of Figure 4B.

“Cerulean-fusion PpREN or PpRopGAP1 (simplified as + REN and + GAP1) were expressed for rescue in the ropgap, ren septuple mutant background.”

Fig 4h: it is not clear to me how the classification of localization and rescuing ability was performed. Please explain.

We apologize for the unclarity. The rescue ability was based on the extent of caulonema cell growth. Protein localization was determined by the relative intensity on the membrane and in the cytosol. Both classifications could be readily and empirically distinguished. We have included our standards for categorization in the legend of **Figure 4H**.

“The rescue capability was categorized as either complete (++), partial (+), or absent (-) when mosses exhibited fully, partially, or no elongated caulonema cells on BCDAT plates, respectively. Protein localization was determined by assessing the relative intensity on the membrane (Mem.) and in the cytosol (Cyto.). If the membrane signals were visually stronger than the cytosolic fluorescence, they were classified as membrane localization.”

Fig 7: If possible, please replace the top-right and bottom-left panels with bright field images having less background particles. It is challenging to assess cell morphology.

As our imaging dishes were not adequately cleaned, we were unable to obtain images with cleaner backgrounds as suggested. Nonetheless, we adjusted the overall brightness and contrast, which significantly improved the recognition of cell morphology.

Reviewer #2 (Remarks to the Author):

In the Manuscript “Two subtypes of GTPase-activating proteins coordinate tip growth and cell size regulation in *Physcomitrium patens*” Ruan et al. show convincingly that two distinct types of GTPase-activating proteins (GAPs) regulate ROP signalling in protonema cells of *Physcomitrium patens*. The exciting and novel aspects of this manuscript are that both types of GAPs act, on the one hand, very redundantly, as only a knockout of all GAPs from both types leads to strong phenotypes. On the other hand, cellular localization and biochemical properties of both GAP types are very distinct and lead to specific modulation of ROP function and, thus, cellular growth. Additionally, the authors describe a novel domain in RopGAPs that is required for ROP binding and clustering of the complex in the plasma membrane, which leads to the clustering of ROP signalling complexes at the plasma membrane.

This manuscript is well-written, convincing and represents original research that is of major significance to the field and our understanding of ROP signalling and the complex formation of signalling components at the plasma membrane. Therefore, this manuscript would be of

major interest to this field. Most experiments and shown results are of high quality, support the authors' conclusions and are sufficiently detailed.

In conclusion, the authors provide a well-written manuscript with convincing results that allow the presented conclusions. Therefore, I suggest the publication of this manuscript in Nature Communications after a few issues are addressed.

We appreciate the reviewer's positive comments and have addressed the mentioned issues below.

Major:

Line 80-82: the authors conclude that RopGAPs, GEFs and ROPs are organized in clusters, which is convincing from the shown images and measurements. However, the authors do not convincingly show that the observed clusters are the very same for all three proteins. As the authors possess lines with RopGAP one coexpressed with either ROP4 or GEF4 this should be included in Figure 2.

We performed colocalization analyses for RopGAP1-RopGEF4 and ROP4-RopGEF4 in membrane particles and found that ~80% of RopGAP1 and ROP4 particles exhibited colocalization with RopGEF4 (Figure 2G). The results are mentioned in the main text as follows.

“In addition, 79% (n = 42 particles from 9 cells) and 80% (n = 44 particles from 9 cells) of PpROP4 and PpRopGAP1 particles, respectively, exhibited colocalization with PpRopGEF4 particles (Figure 2G), implying that they are largely present in the same clusters.”

Line 123-125: The authors use alpha fold to show that RopGAPs have an α -helical domain (PCRIB) preceding the CRIB domain. If this domain is crucial for CRIB folding and function, how is this achieved in other CRIB-domain proteins, like RICs or AtGAP5? The authors should include alpha-fold structures of such proteins, if these proteins may possess a similar fold with a different protein sequence.

This is a very interesting question. We analyzed RICs from *P. patens*, Arabidopsis, rice, and maize (Figure S4A). Consistent with a recent report (Ntefidou et al. Cell Rep. doi: 10.1016/j.celrep.2023.112130), we found that most RICs contain conserved regions before and after the canonical CRIB motifs. Interestingly, the pre-CRIB sequences are different from the PCRIB motifs in RopGAPs, however, they are also predicted to adopt an α -helical conformation (Figure S4B). AtRopGAP5 as well as AtRIC8 does not have a full-length pre-CRIB segment. Thus their N-termini do not form ordered structures. The sequence alignment and predicted structures of representative RICs are shown in Figure S4. The results are mentioned in the main text as follows.

“Interestingly, RICs also contain a conserved region before the CRIB motif (Figure S4A). Although the corresponding sequences are not similar to the PCRIB motifs in RopGAPs, they also tend to form α -helical structures (Figure S4B).”

A major point which needs improvement is the protein-protein interaction analysis. The Y2H results are surprising as one would expect interaction of RopGAP1 not just with the CA-ROP4 but also with the WT version, as active ROPs will also be present in yeast. Furthermore, there is slight growth of yeast in the RopGAP1 – DN-ROP4 combination, suggesting weak interaction between the two. This is also visible in the pull-down experiments, where the authors do not comment on the presence of ROP4 in the WT and H165A versions of RopGAP1. This interaction with inactive ROP might be weak, but it seems stronger than this RenGAP to ROP4. Such an interaction of RopGAP to active and inactive ROPs would explain why RopGAPs behave differently and form clusters with ROPs in active ROP signalling domains. The interaction strength might be different, but it would allow RopGAPs to stay in a complex with ROPs after inactivation and thus would be crucial for persistent complex formation at the plasma membrane. Therefore, it would be crucial to show to what extent RopGAPs interact with DN-ROP and how the interaction strength differs from CA-ROP or RenGAP-ROP interaction. To make such interaction studies convincing and quantitative MST experiments (or similar quantifiable approaches) should be performed. If this is technically not possible, at least the interaction of RopGAPs with all activity forms of ROPs should be compared also with Western blot after the pull-down and compared to the presented RenGAP-ROP interaction.

We performed pull-down and Western blotting assays to compare the interaction of RopGAP1 and REN with ROP4DN, as well as the interaction of RopGAP1 with wild-type ROP4, ROP4CA, and ROP4DN. We confirmed that RopGAP1 preferentially interacted with ROP4CA. However, it also exhibited weaker binding with wild-type ROP4 and ROP4DN. The binding of RopGAP1 with ROP4DN was marginal and was not significantly different from the REN-ROP4DN interaction (Figure S9A and S9C). The results are mentioned in the main text as follows.

“In the pull-down experiments, PpRopGAP1 fused with maltose binding protein (MBP) displayed preferential interactions with Glutathione S-transferase (GST)-tagged PpROP4CA and only weakly interacted with PpROP4 or PpROP4DN (Figure 8C, Figure S9A).”

“These data suggest that PpREN has a much weaker binding ability than PpRopGAP1 toward active PpROP4. However, our western blotting analyses did not reveal differences in the binding ability of PpREN and PpRopGAP1 toward PpROP4DN (Figure S9C).”

To further explain the functional differences between RopGAPs and RenGAPs, we expanded our discussion by comparing possible regulatory mechanisms. The corresponding main text is shown below.

“However, the presence of distinct regulatory domains as well as biochemical characteristics indicates that they may be regulated in different ways. Indeed, the CRIB domain in RopGAPs enhances the GAP activity^{21, 23} and targets RopGAPs to the membrane by interacting with active ROPs (Figure 3). By contrast, the activity of RenGAPs appears to be not influenced by the PH domain or the CC tail domain (Figure S11A)²². An early study reveals a possible role of the CC tail of AtREN1 in association with exocytic vesicles²². Recently, AtPHGAPs have been shown to be stabilized by phosphorylation in the CC tail domain, which consequently inactivates AtROP2 at the indenting regions in pavement cells^{60, 61}. RenGAPs could also potentially associate with the membrane and vesicles via the lipid-binding PH domain²⁶. However, whether this mechanism is widely conserved requires further investigation because the membrane association ability of RenGAPs in moss protonema cells (Figure 4G), pollen tubes²², and root hairs²⁶ seems to be different.”

Minor:

Abstract: The first two sentences should be revisited, as cell polarity is not a mechanism. Cell polarity results from mechanisms leading to concentration differences of cellular components, thus, breaking symmetry.

We have rephrased this sentence as:

“The establishment of cell polarity is a prerequisite for many developmental processes.”

Line 4: “in higher plants”. This should be rephrased to seed plants (spermatophytes). The height of the plants is irrelevant.

We have replaced “higher plants” with “seed plants” as suggested.

Line 17-19: When introducing and discussing ROP clustering at the plasma membrane, the authors should include the work of Alexandre Martinière and Yvon Jaillais. Especially Platre et al. Science 2019 (DOI: 10.1126/science.aav9959) and Smokvarska et al. Current Biology 2020 (DOI:10.1016/j.cub.2020.09.013).

We have included these references in the second paragraph of the Introduction section.

“ROPs have been shown to form membrane clusters in root epidermal cells^{16,17}, pollen tubes¹⁸, and root hairs¹⁹, suggesting the involvement of self-organization in tip-growing cells.”

Line 39: “basal land plants”. The authors should rephrase this. *P. patens* might have a less complex body architecture than other land plants, but it is still a complex organism that evolved for the same time as any other existing organism and is thus not “basal”.

We have rephrased this sentence as:

“The moss Physcomitrium patens (P. patens) represents one of the extant species closest to the land plant ancestor”.

Line 105: Include the construct details (amino acid 1-193, including PCRIB and CRIB domain) also in the main text for easier understanding.

We have included this information as suggested.

“When tagged with mNG, the Nter portion of PpRopGAP1 (amino acid 1-193) was localized to the apical membrane and displayed an enrichment similar to the full-length PpRopGAP1 (Figure 3A). Introducing a histidine-to-alanine mutation in the core CRIB motif (H165A) or removing the Nter portion (Δ 1-193) completely abolished membrane localization”

Line 137: “Expression in vitro”. It is unclear which systems the authors are describing. Was an in vitro translation system used and data is not shown, or do the authors refer to the expression in E. coli?

Yes, we meant the expression in E. coli. We have included this information in the main text as follows.

“Moreover, we failed to express R134/135L mutant proteins using bacterial expression systems in vitro.”

Line 162: “expression is completely blocked”. The authors do not test expression (mRNA production and presence) and should thus rephrase this.

We have rephrased this sentence as:

“PpRopGAP expression is substantially blocked at the protein level”.

Line 191: “local inactivation of ROPs by RopGAPs is required”. It is unclear why one can conclude that local inactivation is required. It might be that the overall reduction of RopGAP activity causes the mild phenotypes of the ropgap mutant. The authors should clarify this.

We agree with the reviewer that the overall reduction of RopGAP activity can also contribute to the mutant phenotypes because the CRIB domain not only targets RopGAPs to the membrane but also enhances the GAP activity. Since it is technically difficult to distinguish these two functions, we deleted the words “is required” and concluded that:

“local inactivation of ROPs by RopGAPs is not essential for polarity formation and tip growth.”

Line 199: “In ropgap or ren mutants alone”. Unclear reference to genotype. Rephrase if single mutants or which genotype is meant.

We meant the ropgap sextuple mutant and ren mutant. The text has been modified as follows:

“In the ropgap sextuple mutant or ren mutant alone, ...”.

Line 230: “we frequently observed early accumulation“ should be rephrased, as this was observed in all cells.

We deleted “frequently”.

Line 361: “found exclusively in land plants and K. nitens”. The authors should consider that our knowledge about the genomes of other streptophyte algae is very limited. If RopGAPs and RenGAPs are found in k. nitens, it is very likely that they will also be present in other Streptophyte algae and thus have an earlier evolutionary origin than suggested by the authors.

We have rephrased this sentence as:

“...they have emerged around the time of the colonization of land by plants or earlier”.

REVIEWERS' COMMENTS

Reviewer #1 (Remarks to the Author):

I thank the authors for addressing all my comments satisfactorily. I now fully support publication.

Reviewer #2 (Remarks to the Author):

The revised version of the Manuscript „Two subtypes of GTPase-activating proteins coordinate tip growth and cell size regulation in *Physcomitrium patens*” by Ruan et al. improved over the initial version and addressed all comments and requests. The authors show convincingly that two distinct types of GTPase-activating proteins (GAPs) regulate ROP signalling in protonema cells of *Physcomitrium patens*. Both GAPs types act very redundant, but cell biological and biochemical properties of both GAP types are very distinct and lead to specific modulation ROPs and cellular growth. RopGAPs colocalize with ROPs and GEFs at the plasma membrane, bind active ROPs strongly and even inactive ROPs weakly, and only show weak GAP activity. REN/PHGAPs do not show plasma membrane association and bind ROPs very weakly, but show a high GAP activity. Additionally, a novel α -helical domain in RopGAPs is described that is required for ROP binding and complex clustering at the plasma membrane. Such a structural component adjacent to the CRIB domain might also be present in RIC proteins.

This manuscript improved over the last version and is well-written, and convincing, with high-quality figures and appropriate conclusions. All comments and requests were addressed and supported with new analyses or experiments. Also, all changes to the manuscript text were addressed in full. The presented data is a significant contribution to the general field and especially to the understanding of ROP signalling complex activity at the plasma membrane. Therefore, this manuscript would be of major interest to this field.

In conclusion, the authors provide a well-written manuscript with convincing results that allow the presented conclusions. Therefore, I suggest the publication of this manuscript in Nature Communications.